# Rtf1-dependent transcriptional pausing regulates cardiogenesis

Adam D Langenbacher, Fei Lu, Luna Tsang, Zi Yi Stephanie Huang, Benjamin Keer, Zhiyu Tian, Alette Eide, Matteo Pellegrini, Haruko Nakano, Atsushi Nakano, Jau-Nian Chen*

Department of Molecular, Cell, and Developmental Biology, University of California, Los Angeles, Los Angeles, United States

## eLife Assessment

This **important** study shows that a controlled pause in gene reading is required for early heart cells to form during development. The authors demonstrate that loss of this pause prevents the proper activation of the heart-producing program across animal and stem cell systems. The evidence is **compelling**, supported by careful genomic and functional analyses that clearly define the developmental block. Overall, this work will interest developmental biologists and inspire further studies on the origins of early heart defects.

*For correspondence:
chenjn@g.ucla.edu

**Competing interest:** The authors declare that no competing interests exist.

**Abstract** Transcriptional pause-release critically regulates cellular RNA biogenesis, yet how dysregulation of this process impacts embryonic development is not fully understood. Rtf1 is a multifunctional transcription regulatory protein involved in modulating promoter-proximal pausing of RNA Polymerase II (RNA Pol II). Using zebrafish and mouse as model systems, we show that Rtf1 activity is essential for the differentiation of the myocardial lineage from mesoderm. Ablation of *rtf1* impairs the formation of *nkx2.5+/tbx5a+* cardiac progenitor cells, resulting in the development of embryos without cardiomyocytes. Structure-function analysis demonstrates that Rtf1's cardiogenic activity requires its Plus3 domain, which confers interaction with the pausing/elongation factor Spt5. In Rtf1-deficient embryos, the occupancy of RNA Pol II at transcription start sites was reduced relative to downstream occupancy, suggesting a reduction in transcriptional pausing. Intriguingly, attenuating pause release by pharmacological inhibition or morpholino targeting of CDK9 improved RNA Pol II occupancy at the transcription start sites of key cardiac genes and restored cardiomyocytes in Rtf1-deficient embryos. Thus, our findings demonstrate the crucial role that Rtf1-mediated transcriptional pausing plays in controlling the precise spatiotemporal transcription programs that govern early heart development.

## Introduction

Cardiogenesis relies on tightly controlled transcriptional programs regulated at all stages of mRNA synthesis. Decades of studies have identified an evolutionarily conserved network of transcription factors that direct embryonic heart development. Notably, members of the GATA, NK-2 homeobox, TBX, HAND, and MEF2 transcription factor families play crucial, conserved, and well-characterized roles in specifying cardiac cells from the mesoderm (*Bruneau, 2013*; *Paige et al., 2015*; *Rana et al., 2013*; *Lu et al., 2016*). Through a careful balance of positive and negative regulatory signals, the cardiac transcription regulatory network coordinates with other transcriptional networks to ensure that the appropriate number of cardiac progenitors is specified. In zebrafish, this activity partitions the anterior lateral plate mesoderm (ALPM) into multiple fates, including the cardiac mesoderm, which is

bounded anteriorly by blood and endothelial progenitors and posteriorly by pectoral fin progenitors (*Schoenebeck et al., 2007*; *Mao et al., 2021*; *Duong et al., 2021*; *Sorrell and Waxman, 2011*).

Along with the paradigmatic role played by cardiac transcription factors in transcriptional initiation, accumulating evidence suggests that transcriptional control of cardiogenesis is also accomplished by epigenetic and post-initiation mechanisms. For example, the Brg1/Brm-associated factor (BAF) complex, an ATP-dependent chromatin remodeling complex that actively positions nucleosomes, is critical for normal vertebrate cardiogenesis (*Hota et al., 2019*; *Takeuchi et al., 2011*; *Lou et al., 2011*). The BAF complex directly interacts with cardiac transcription factors including Tbx5, Nkx2-5, and Gata4 (*Lickert et al., 2004*) and controls temporal changes in chromatin accessibility during cardiac differentiation (*Hota et al., 2019*). Intriguingly, forced expression of the BAF component Baf60c along with Tbx5/Nkx2-5/Gata4 can induce cardiac differentiation from non-cardiac mesoderm (*Takeuchi and Bruneau, 2009*), suggesting that nucleosomal positioning and chromatin accessibility are key mechanisms regulating the activity of cardiac transcription factors during cardiogenesis.

Cardiac gene expression is also sensitive to transcription elongation regulation. Spt6 (Suppressor of Ty 6) is a histone chaperone that promotes RNA Polymerase II (RNA Pol II) elongation and co-transcriptionally modulates chromatin structure (*Endoh et al., 2004*; *Hartzog et al., 1998*). In zebrafish, Spt6 interacts genetically with the transcriptional pausing/elongation factor Spt5 (Suppressor of Ty 5) to support Nkx2.5+ cardiac progenitor specification (*Keegan et al., 2002*). While both Spt6 and Spt5 promote transcription elongation, they likely do so via distinct mechanisms. Spt5 directly interacts with Spt4 to form the DRB sensitivity-inducing factor (DSIF) complex, which has both inhibitory and stimulatory roles in transcription (*Wada et al., 1998*). Unphosphorylated Spt5 fosters promoter-proximal pausing of RNA Pol II, but phosphorylation of Spt5's C-terminal repeat region by cyclin-dependent kinase 9 (Cdk9) promotes transcriptional pause release and converts DSIF into an elongation-promoting factor (*Wier et al., 2013*; *Chen et al., 2020*; *Hu et al., 2021*). How the transcription regulatory activities of these elongation and pausing factors facilitate cardiac development is not yet understood.

The Polymerase Associated Factor 1 Complex (PAF1C) consists of five core proteins (Paf1, Ctr9, Cdc73, Rtf1, and Leo1) and plays wide-ranging transcription regulatory roles, including promoting co-transcriptional epigenetic modification of histones, supporting both pausing and elongation of RNA Pol II, influencing mRNA 3' end formation, and serving as a bridge between specific transcription factors and RNA Pol II (*Park et al., 2023*; *Van Oss et al., 2017*; *Jaehning, 2010*). In zebrafish, PAF1C subunits differentially regulate cardiac development (*Langenbacher et al., 2011*). Leo1 activity is dispensable for the formation of cardiac progenitors, but is required for the differentiation of the cardiac chambers (*Nguyen et al., 2010*). Embryos lacking Ctr9, Cdc73, or Paf1, on the other hand, exhibit a deficiency of cardiomyocytes and only form small, dysmorphic hearts. Intriguingly, loss of the Rtf1 subunit of the PAF1C results in the most dramatic defects in cardiogenesis. Rtf1-deficient embryos have a severe reduction in *nkx2.5+* cardiac progenitors and fail to generate any cardiomyocytes (*Langenbacher et al., 2011*).

Rtf1 is a multifunctional transcription regulatory protein that modulates pausing and elongation of RNA Pol II, as well as histone epigenetic modifications. Given Rtf1's numerous roles in transcription regulation, it remains an open question how Rtf1 mechanistically regulates cardiogenesis. In this study, we showed that zygotic *rtf1* mutants, like *rtf1* morphants, lack expression of most cardiac progenitor marker genes and produce no cardiomyocytes in zebrafish. We also found that knockout of Rtf1 in the cardiogenic mesoderm of mouse embryos and knockdown of Rtf1 in mouse embryonic stem cells (mESCs) reduced the formation of cardiac tissue, indicating that the role of Rtf1 in specification of the cardiac lineage is conserved among vertebrates. Using FACS and transcriptomic profiling approaches, we show that differentiation of the lateral plate mesoderm (LPM) into cardiac progenitors requires Rtf1 activity. We further show that Rtf1's Plus3 domain is essential for cardiac progenitor formation, while its histone modification domain (HMD) is dispensable. Consistent with the Plus3 domain's function as an interaction point between Rtf1 and the DSIF component Spt5, Rtf1-deficient embryos display a genome-wide decrease in promoter-proximal pausing of RNA Pol II. Excitingly, attenuating pause release by chemical and morpholino inhibition of CDK9 improved pausing at cardiac genes and restored cardiac progenitor formation to Rtf1-deficient embryos. Together, our findings suggest that during development, Rtf1 facilitates expression of the cardiac transcription program by supporting the promoter-proximal pausing of RNA Pol II.

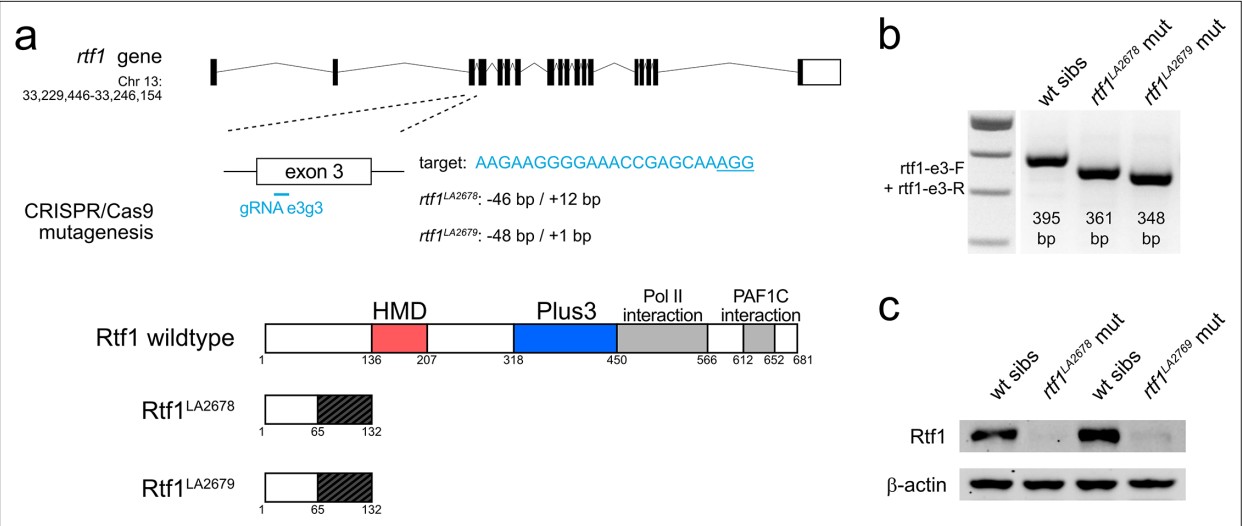

**Figure 1.** Generation of zebrafish *rtf1* null mutants with CRISPR/Cas9. (**a**) Schematic of CRISPR/Cas9 mutagenesis of the *rtf1* gene. Two mutant alleles, *rtf1*[LA2678] and *rtf1*[LA2679], were recovered after targeting of *rtf1* exon 3. Both alleles are predicted to disrupt translation of the Rtf1 protein and eliminate the histone modification domain (HMD), Plus3, polymerase II (Pol II) interaction, and Polymerase Associated Factor 1 Complex (PAF1C) interaction domains. (**b**) Agarose gel electrophoresis results of genotyping *rtf1* mutants by PCR. Deletions in *rtf1*[LA2678] and *rtf1*[LA2679] alleles can be distinguished from wild-type allele using primers rtf1-e3-F and rtf1-e3-R. (**c**) Western blot detecting Rtf1 and β-actin (loading control) proteins in lysates of wild-type and *rtf1* mutant embryos. The image is representative of two independent experiments.

The online version of this article includes the following source data for figure 1:

**Source data 1.** PDF file containing original agarose gel image for *Figure 1b*, indicating the relevant bands and genotypes.

**Source data 2.** Original file for agarose gel image displayed in *Figure 1b*.

**Source data 3.** PDF file containing original western blots for *Figure 1c*, indicating the relevant bands and genotypes.

**Source data 4.** Original files for western blots displayed in *Figure 1c*.

## Results

### Rtf1 is essential for heart development in zebrafish

We previously showed that knockdown of *rtf1* abolishes cardiac progenitor cell formation in zebrafish. To corroborate this finding, we used CRISPR/Cas9 to generate stable *rtf1* mutant lines. We identified two independent lines (*rtf1*[LA2678] and *rtf1*[LA2679]) with small deletions in exon 3 of the *rtf1* gene that are predicted to result in frameshifts and early stop codons, which nearly eliminate the production of Rtf1 protein (*Figure 1a–c*). Like *rtf1* morphants and the previously reported *rtf1*[KT641] allele (*Langenbacher et al., 2011*; *Akanuma et al., 2007*), both *rtf1*[LA2678] and *rtf1*[LA2679] mutants have poorly defined posterior somite boundaries but are otherwise visually indistinguishable from their wild-type siblings during early somitogenesis, the developmental timeframe when cardiac progenitors emerge from the LPM. After 1 day of development, cardiac tissue is not detectable in *rtf1* mutant embryos by visual inspection or by *myl7* expression (*Figure 2a and b*) and mutant embryos display a downwardly curved body shape and an absence of pigmentation at 2 days post-fertilization (*Figure 2—figure supplement 1*). In situ hybridization further showed that *rtf1*[LA2678] and *rtf1*[LA2679] mutant embryos, like *rtf1* morphants, fail to express normal levels of critical cardiac transcription factors, including *mef2ca, nkx2.5, tbx5a, and txb20* at the time when cardiac progenitors emerge from the ALPM (*Figure 2c–j*; *Langenbacher et al., 2011*; *Lu et al., 2017*). Together, these data show that morpholino knockdown faithfully recapitulates the *rtf1* mutant phenotype (*Langenbacher et al., 2011*) and demonstrate that Rtf1 activity is required for the expression of multiple cardiac transcription factors, indicating a critical role of Rtf1 in regulating the cardiac gene program.

### Conserved role of Rtf1 in mammalian cardiac development

The sequence of the Rtf1 protein is highly conserved among vertebrates (83% identity at the amino acid level between fish and humans). To examine if the crucial role of Rtf1 in cardiogenesis we observed in zebrafish was conserved in mammals, we expressed Cre-recombinase under the control

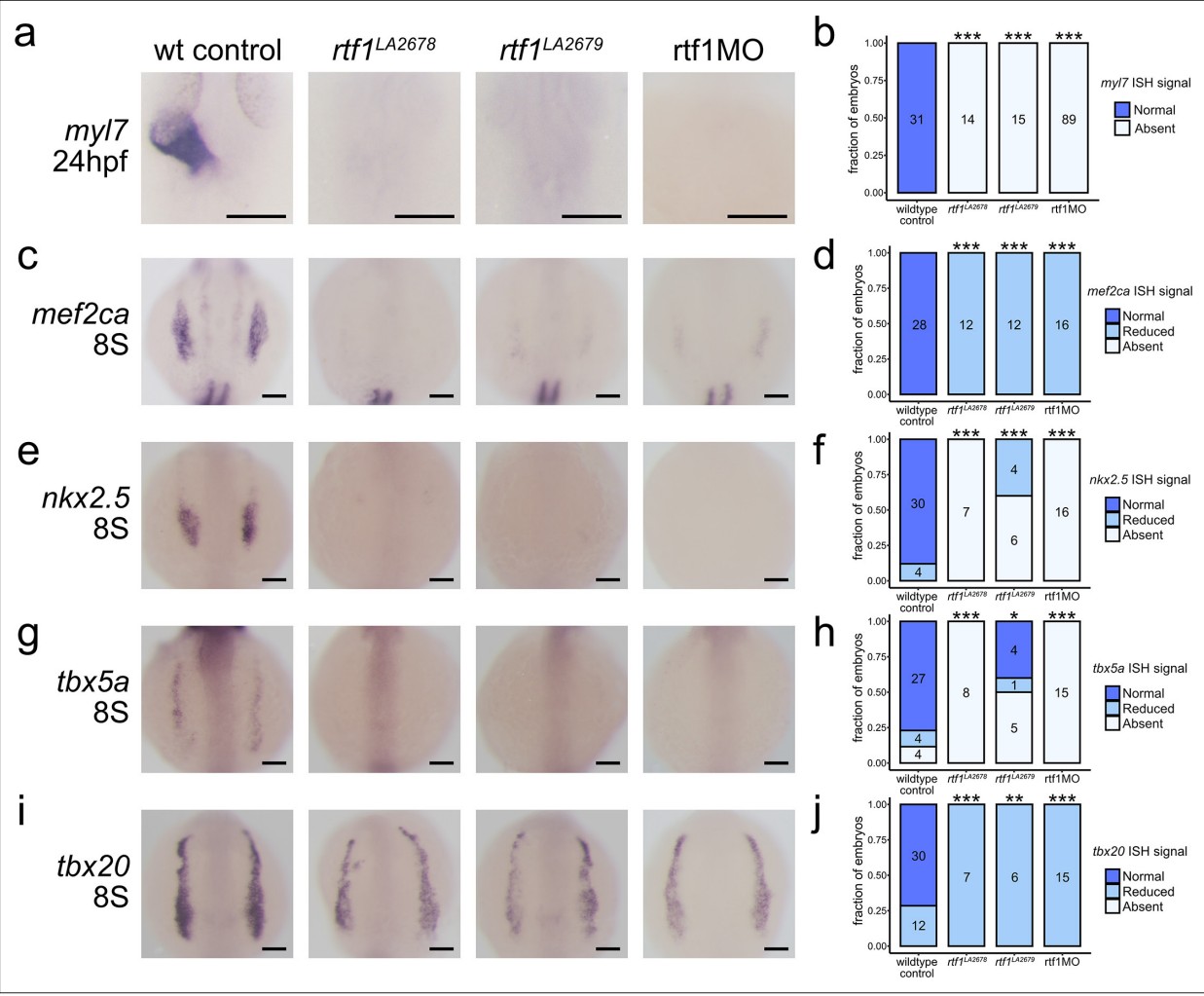

**Figure 2.** Characterization of early cardiac development in Rtf1-deficient embryos. (**a**) Representative images of RNA in situ hybridization detecting *myl7* expression in 24 hpf zebrafish embryos. (**b**) Quantification of *myl7* signal intensity in control and Rtf1-deficient embryos at 24 hpf. Numbers on bars indicate the number of embryos analyzed. (**c**) Representative images of RNA in situ hybridization detecting *mef2ca* expression in 8 somite stage zebrafish embryos. (**d**) Quantification of *mef2ca* signal intensity in control and Rtf1-deficient embryos at the 8 somite stage. Numbers on bars indicate the number of embryos analyzed. (**e**) Representative images of RNA in situ hybridization detecting *nkx2.5* expression in 8 somite stage zebrafish embryos. (**f**) Quantification of *nkx2.5* signal intensity in control and Rtf1-deficient embryos at the 8 somite stage. Numbers on bars indicate the number of embryos analyzed. (**g**) Representative images of RNA in situ hybridization detecting *tbx5a* expression in 8 somite stage zebrafish embryos. (**h**) Quantification of *tbx5a* signal intensity in control and Rtf1-deficient embryos at the 8 somite stage. Numbers on bars indicate the number of embryos analyzed. (**i**) Representative images of RNA in situ hybridization detecting *tbx20* expression in 8 somite stage zebrafish embryos. (**j**) Quantification of *tbx20* signal intensity in control and Rtf1-deficient embryos at the 8 somite stage. Numbers on bars indicate the number of embryos analyzed. Scale bars in a, c, e, g, and i represent 0.1 mm. *: $p < 0.05$, **: $p < 0.01$, ***: $p < 0.001$ based on Fisher's exact test.

The online version of this article includes the following figure supplement(s) for figure 2:

**Figure supplement 1.** Bright-field microscopic images of 2 days post-fertilization (d2) wild-type (**a**) and *rtf1*^LA2679^ mutant (**b**) embryos.

of the *Mesp1* promoter in *Rtf1* flox mouse embryos (***Figure 3a***; ***Langenbacher et al., 2023***). The *Mesp1* promoter drives expression of Cre in the early cardiogenic mesoderm (E6.25) (***Saga et al., 1999***), allowing for knockout of Rtf1 activity prior to the onset of heart development. *Rtf1 Mesp1*-Cre knockout mouse embryos die prior to birth, suggesting that Rtf1 function in the cardiogenic mesoderm is essential for survival. We examined the developing hearts of *Rtf1 Mesp1*-Cre knockouts and their wild-type siblings by in situ hybridization and found that while Cre+;*Rtf1*^flox/+^ and Cre- embryos display an *Nkx2.5*+/*Tbx20*+ heart tube undergoing looping at E8.5, *Rtf1 Mesp1*-Cre knockouts have little or no expression of these markers (***Figure 3b and c***). These data suggest a conservation of the critical role of Rtf1 in differentiation of the myocardial lineage from the mesoderm.

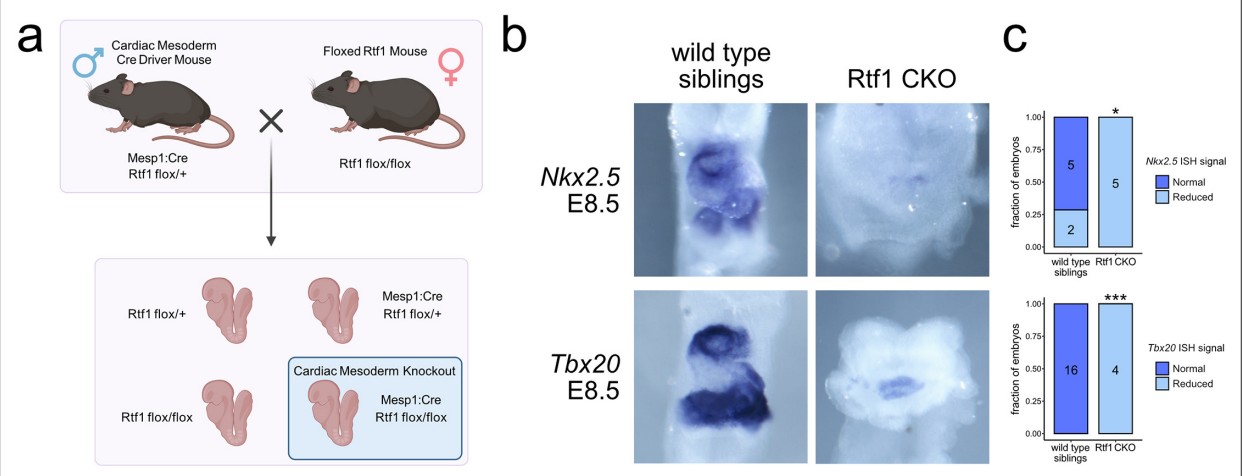

**Figure 3.** Failure of cardiac differentiation upon loss of Rtf1 in the mammalian cardiac mesoderm. (**a**) Diagram of generation of cardiac mesoderm-specific Rtf1 knockout mouse embryos. A *Mesp1*:Cre insertion allele was bred into an *Rtf1* exon 3 floxed background. *Rtf1*flox/+ mice with *Mesp1*Cre/+ were bred to homozygous *Rtf1*flox/flox females, resulting in one quarter of embryos lacking Rtf1 activity in the cardiac mesoderm (Rtf1 CKO). (**b**) Representative images of RNA in situ hybridization detecting *Nkx2.5* (wild type n=7, Rtf1 CKO n=5) and *Tbx20* (wild type n=16, Rtf1 CKO n=4) gene expression in Cre-negative (wild-type siblings) and Cre-positive *Rtf1*flox/flox (Rtf1 CKO) embryos at E8.5. (**c**) Quantification of *Nkx2.5* and *Tbx20* signal intensity in Cre-negative (wild-type siblings) and Cre-positive *Rtf1*flox/flox (Rtf1 CKO) embryos at E8.5. Numbers on bars indicate the number of embryos analyzed. *: p<0.05, ***: p<0.001 based on Fisher's exact test.(**a**) created with BioRender.com.

To further investigate the conservation of Rtf1's role in cardiac differentiation, we knocked down Rtf1 activity in mESCs using short hairpin RNA (shRNA) (*Figure 4a and b*). Unmanipulated embryoid bodies (EBs) derived from mESCs are capable of differentiating into cardiac tissues and form beating patches (*Figure 4c and d*). In sharp contrast, Rtf1-deficient mESCs display a dramatic reduction in *Myh6*, *Nkx2.5*, and *Nppa* expression and produce significantly fewer beating patches (~67% reduction) (*Figure 4a–d*). Interestingly, the temporally dynamic expression levels of *Brachyury* (mesoderm), *Afp* (endoderm), and *Ncam1* (ectoderm) are not significantly affected by Rtf1 knockdown (*Figure 4e–g*). Together, these data show that Rtf1 is dispensable for the formation of the three germ layers but is essential for differentiation of the cardiac lineage from mesoderm in mammals. This phenotype is similar to what we observed in zebrafish embryos, indicating a conservation of Rtf1's function in cardiogenesis between fish and mammals.

## Rtf1 is required for expression of genes associated with differentiation of cardiac precursors from the lateral plate mesoderm

To gain insight into how Rtf1 deficiency impacts differentiation of the cardiac lineage, we profiled LPM transcriptomes of control and *rtf1* morphant embryos. We previously showed that loss of Rtf1 activity does not disrupt expression of the LPM marker *hand2* (*Langenbacher et al., 2011*). We therefore took advantage of a hand2:GFP zebrafish transgenic reporter line to isolate LPM cells from wild-type and Rtf1-deficient embryos (*Figure 5a–c*, *Figure 5—figure supplement 1*). Compared to wild-type control samples, the *rtf1* morphant LPM displayed 1420 significantly downregulated and 1185 significantly upregulated genes (adjusted p-value<0.05) at the 10–12 somite stage when cardiac progenitor formation is underway in the LPM (*Source data 1*). We examined the most significantly differentially expressed genes and observed that many of the significantly downregulated genes, including *wnt2bb*, *mef2cb*, *mef2ca*, *isl2b*, *tnnt2a*, *atp2a2a*, *rbfox1l*, and *nkx2.5*, are cardiac-lineage markers or have critical known roles in cardiac development (*Figure 5d*). We further found that the most significantly downregulated genes were highly enriched for gene ontologies related to cardiac development, embryonic heart morphogenesis, and muscle cell development (*Figure 5e*). These data show that Rtf1 activity is required for expression of a broad cardiac gene program in the LPM that is likely under the control of multiple cardiac transcription factors.

To further analyze the deployment of the cardiac gene program during cardiac progenitor formation, we performed Single Cell Multiome sequencing of control and *rtf1* morphant embryos.

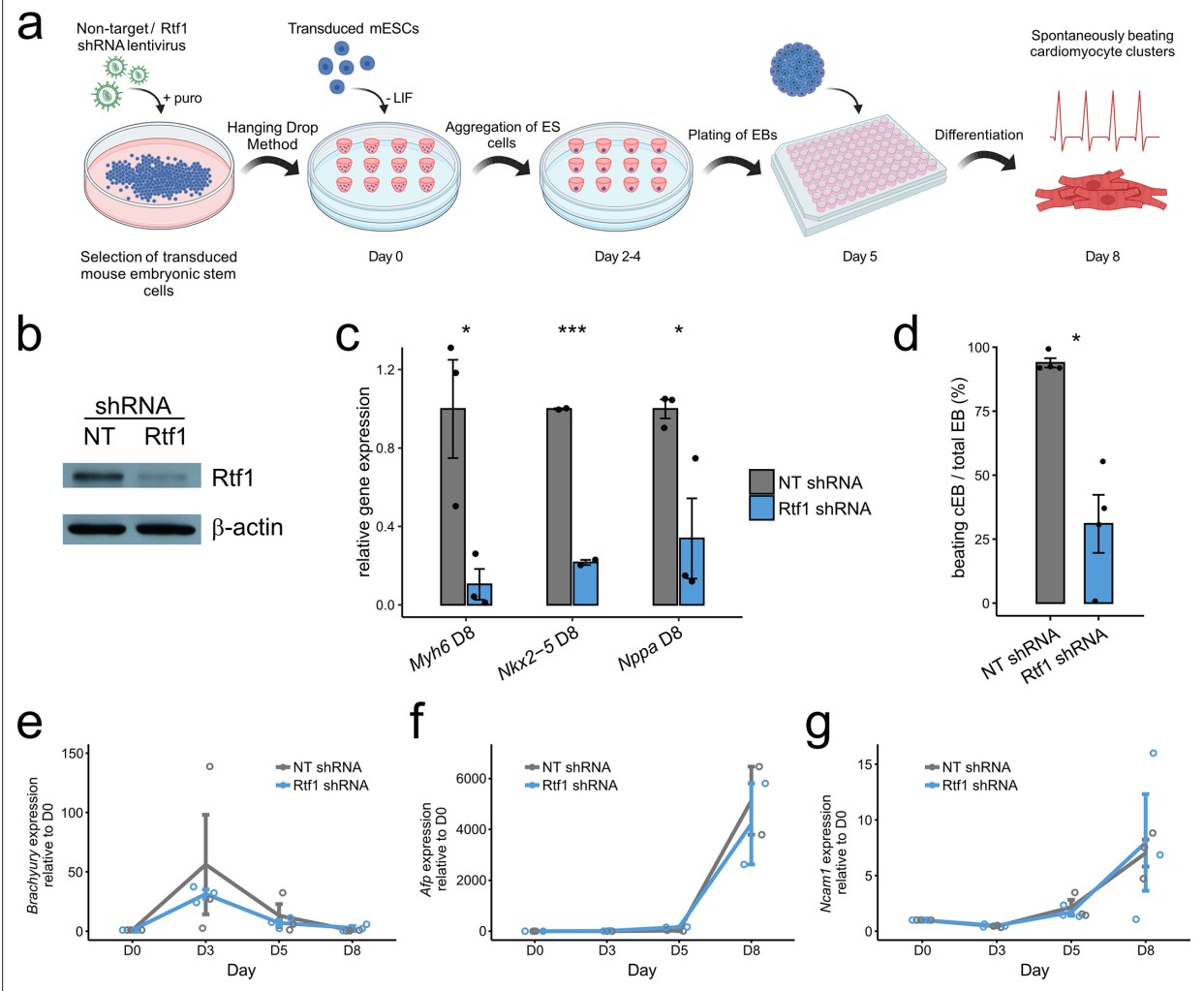

**Figure 4.** Failure of cardiac differentiation from Rtf1 knockdown mouse embryonic stem cells (mESCs). (**a**) Diagram of short hairpin RNA (shRNA) knockdown of Rtf1 in mESCs and differentiation of plated embryoid bodies (EBs). mESCs were transduced with non-target control or Rtf1 shRNA lentivirus and selected with puromycin. Transduced mESCs were grown into EBs using the hanging drop method, which were then plated to examine differentiation into cell types, including beating cardiomyocyte clusters. (**b**) Western blot verifying reduction of Rtf1 protein in Rtf1 shRNA mESCs (~70% reduced based on densitometry) compared to unchanged level of the loading control protein β-actin. Image is representative of three independent experiments. (**c**) qPCR analysis of *Myh6*, *Nkx2-5*, and *Nppa* expression in non-target control (NT) and Rtf1 shRNA plated EBs. Data are normalized to the mean expression level in NT shRNA samples, and error bars indicate the standard error of the mean. (**d**) Mean percentage (± standard error) of plated EBs exhibiting beating cardiomyocyte differentiation (cEBs) in NT and Rtf1 shRNA plated EBs. (**e**) qPCR analysis of *Brachyury* expression in non-target control (NT) and Rtf1 shRNA plated EBs. (**f**) qPCR analysis of *Afp* expression in non-target control (NT) and Rtf1 shRNA plated EBs. (**g**) qPCR analysis of *Ncam1* expression in non-target control (NT) and Rtf1 shRNA plated EBs. Data in e, f, and g are normalized to the mean expression level in NT shRNA samples at day 0 (**D0**), and error bars indicate the standard error of the mean. *: p<0.05, ***: p<0.001. (**a**) created with BioRender.com.

The online version of this article includes the following source data for figure 4:

**Source data 1.** PDF file containing original western blot for *Figure 4b*, indicating the relevant bands and treatments.

**Source data 2.** Original file for the western blot displayed in *Figure 4b*.

Clustering of control and *rtf1* morphant cells based on gene expression and chromatin accessibility resolved 39 cell types corresponding to expected major cell lineages (*Figure 6—figure supplement 1*, *Figure 6—figure supplement 2*, *Supplementary file 2*, *Source data 2*). Interestingly, while *rtf1* morphant cells of each cell type were present, the frequencies of cell types differed significantly between controls and *rtf1* morphants (*Figure 6—figure supplement 3*). Notably, LPM cells are significantly more abundant in *rtf1* morphants, and ALPM derivative cell types (putative cardiac mesoderm,

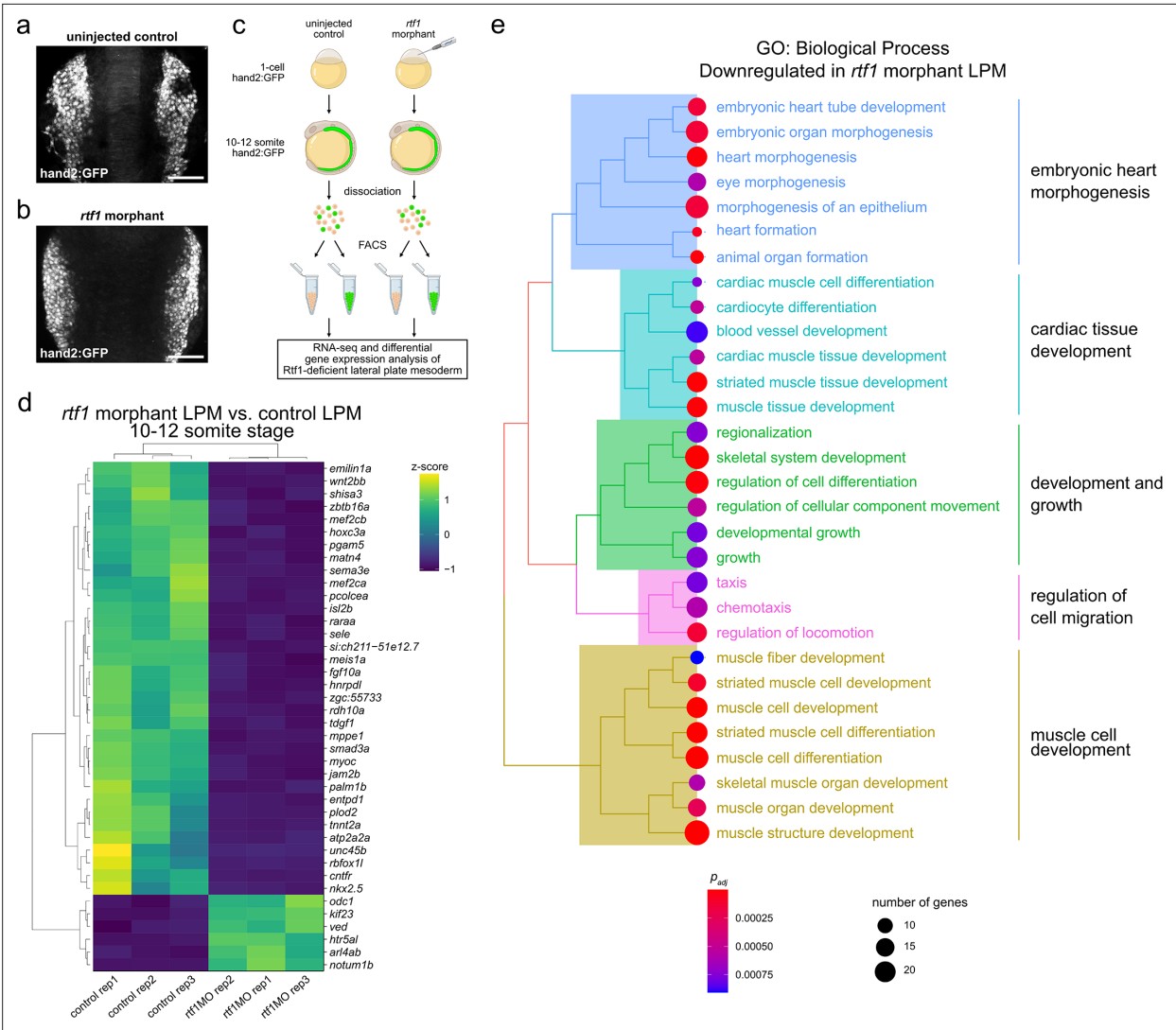

**Figure 5.** Dysregulation of cardiac gene expression in the Rtf1-deficient lateral plate mesoderm. (**a,b**) Projections of confocal z-stacks of hand2:GFP signal in 11 somite stage uninjected control (**a**) and *rtf1* morphant (**b**) embryos. Scale bars represent 100 µm. (**c**) Diagram of lateral plate mesoderm (LPM) FACS RNA-seq experiment. Transgenic hand2:GFP embryos were injected at the one-cell stage with *rtf1* morpholino and grown to the 10–12 somite stage prior to dissociation and sorting based on GFP expression. RNA isolated from GFP-positive LPM cells was subjected to RNA-seq and compared to RNA from uninjected embryo LPM cells. (**d**) Heatmap of scaled expression levels of the top 40 genes most significantly differentially expressed (based on p-value) between uninjected control and *rtf1* morphant hand2:GFP-positive LPM. High and low z-scores represent high and low gene expression, respectively. (**e**) Hierarchical clustering of Biological Process gene ontology (GO) terms based on semantic similarity for the genes most significantly downregulated in the *rtf1* morphant hand2:GFP-positive LPM. (**c**) created with BioRender.com.

The online version of this article includes the following figure supplement(s) for figure 5:

**Figure supplement 1.** Gating strategy for 10–12 somite stage zebrafish hand2:GFP FACS.

endothelial precursors, and rostral blood precursors) are less abundant, indicating that Rtf1 deficiency negatively impacts ALPM differentiation (***Figure 6—figure supplement 3***).

We next examined developmental trajectories within the ALPM by re-clustering ALPM cells (*sema3e+*/*gata5*+ LPM) and ALPM derivatives (***Figure 6a and b***). By calculating pseudotime trajectories, we observed a gradual increase in the expression levels of cardiac markers, including *ttn.2*, *mef2cb*, *tnnt2a*, *ryr2b*, and *myh7bb* along the ALPM-cardiac developmental trajectory in control embryos (***Figure 6c–e***). In *rtf1* morphants, the proportion of cells representing undifferentiated ALPM was increased (37% in *rtf1* morphants vs. 15.5% in controls, ***Figure 6b***, ***Figure 6—figure supplement 4***). Some cells (26.6%) did appear to differentiate along the cardiac trajectory in *rtf1* morphants; like control cardiac progenitors, they maintained the expression of *gata5*, downregulated expression

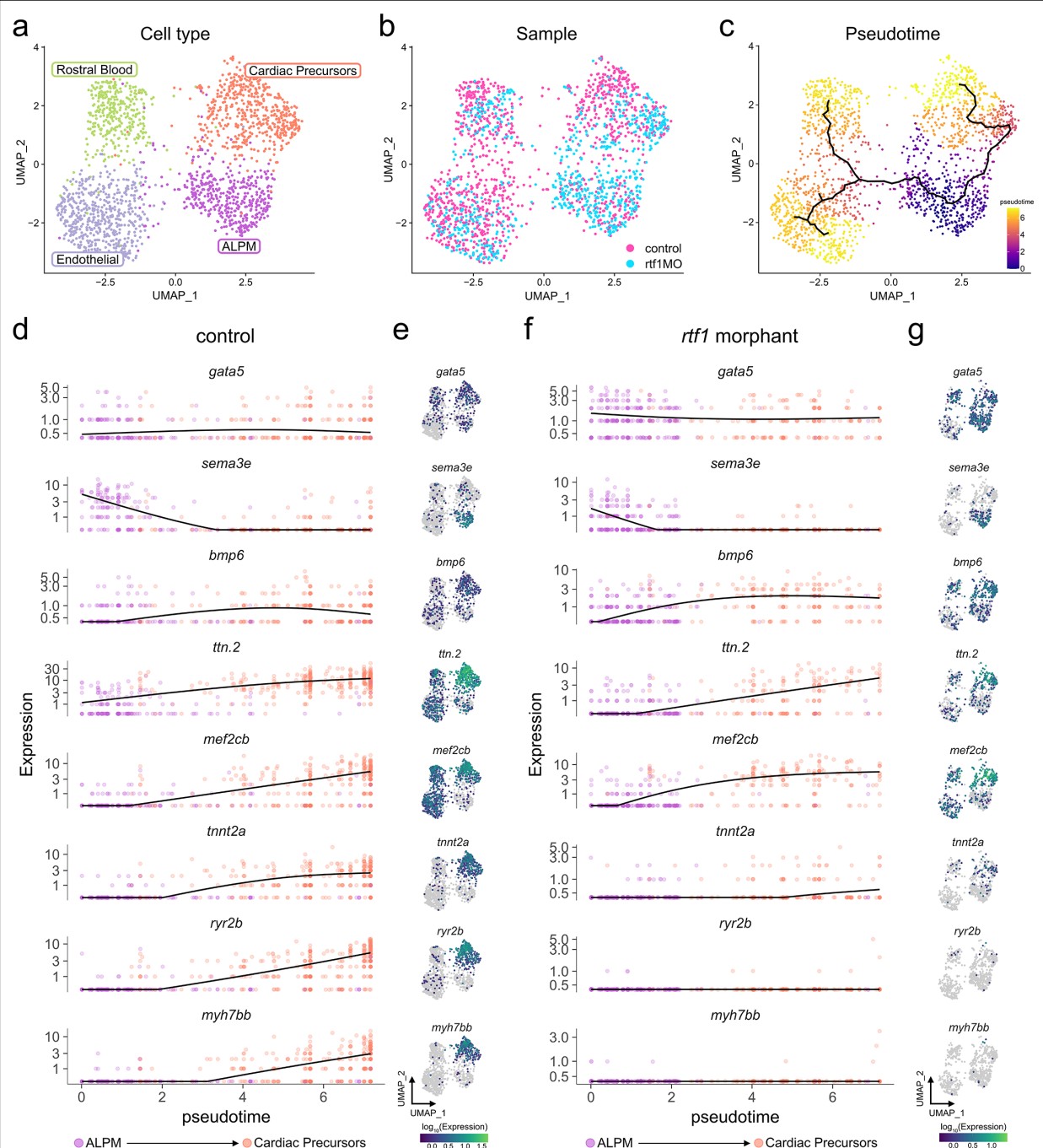

**Figure 6.** Rtf1 is required for transcriptional progression of cardiac precursors. (**a–c**) UMAP plots of anterior lateral plate mesoderm (ALPM) and derivatives from merged and integrated single-cell RNA-seq datasets of 11–12 somite stage uninjected control and *rtf1* morphant embryos. Cells in plots are colored by cell type (**a**), sample (**b**), and pseudotime (**c**). (**d,f**) Expression dynamics plots displaying expression levels (y-axis) for selected genes in uninjected control (**d**) and *rtf1* morphant (**f**) cells over the pseudotime trajectory (x-axis) from ALPM (root) to cardiac precursor state shown in (**c**). (**e,g**) UMAP plots of uninjected control (**e**) and *rtf1* morphant (**g**) cells colored by gene expression levels.

The online version of this article includes the following figure supplement(s) for figure 6:

**Figure supplement 1.** UMAP plots of 11–12 somite stage control and *rtf1* morphant embryo integrated multimodal (ATAC+Gene Expression) single-cell sequencing datasets split based on sample (control on left vs. *rtf1* morphant on right).

**Figure supplement 2.** Clustered dot plot displaying expression of marker genes in cells belonging to each Seurat cluster identified in the integrated multimodal single-cell sequencing analysis of 11–12 somite stage control and *rtf1* morphant embryos.

of the ALPM marker *sema3e*, and initiated expression of the early cardiac lineage genes *bmp6*, *ttn.2*, and *mef2cb* (*Figure 6d–g*, *Figure 6—figure supplement 4*, *Figure 6—figure supplement 5*). However, loss of Rtf1 activity prevented the proper expression of cardiac lineage-restricted markers that are normally activated later in the ALPM-cardiac trajectory, including *tnnt2a*, *ryr2b*, and *myh7bb* (*Figure 6d–g*, *Figure 6—figure supplement 4*, *Figure 6—figure supplement 5*). Taken together, these data suggest that Rtf1 activity is required for gene expression changes associated with proper cardiac progenitor differentiation.

## Rtf1's Plus3 functional domain is necessary for cardiac progenitor formation

The Rtf1 protein is a multifunctional platform for transcription regulation: it consists of several defined functional domains that mediate its interaction with other transcription regulatory proteins and serve as additional points of contact for these proteins with the transcription apparatus (*Figures 1a and 7a*; *Warner et al., 2007*; *Vos et al., 2020*). The N-terminal half of Rtf1 notably contains its HMD, which mediates its interaction with E2 ubiquitin-conjugating enzymes and supports the ubiquitylation of histone H2B K120 (*Cucinotta et al., 2019*; *Van Oss et al., 2016*; *Piro et al., 2012*). Rtf1's C-terminus contains the Plus3 domain, which interacts with the C-terminal repeat region of the pausing and elongation factor Spt5 (*Wier et al., 2013*; *Chen et al., 2020*; *de Jong et al., 2008*), a Pol II interaction domain that allosterically stimulates transcription elongation (*Vos et al., 2020*) and a PAF1C interaction domain.

We investigated if Rtf1 could function in a modular fashion, and if a specific domain or domains were required for the formation of cardiac progenitors during development. To this end, we generated several constructs to express Flag-tagged wild-type and mutant versions of Rtf1 lacking specific functional domains and tested whether they could support cardiogenesis in an *rtf1* morphant background. These constructs also carry 8 silent mutations that alter the *rtf1* translation blocking morpholino binding site to ensure successful translation of synthesized mRNA. Wild-type (Rtf1 wt*) and mutant FLAG-tagged Rtf1 proteins lacking the HMD (ΔHMD) or Plus3 domain (ΔPlus3) were expressed at comparable levels and properly localized to the nucleus, demonstrating that the activity of these proteins was not compromised by instability or mislocalization (*Figure 7b*). Injection of Rtf1 wt* mRNA into *rtf1* morphant embryos was able to robustly rescue the knockdown phenotype and support normal formation of the embryonic heart tube (*Figure 7c*). Interestingly, we found that Rtf1's HMD was dispensable for its role in cardiac progenitor formation (100% of Rtf1 wt* injected *rtf1* morphants with *nkx2.5* signal, n=25 vs. 100% of Rtf1 ΔHMD *rtf1* morphants with *nkx2.5* signal, n=25) (*Figure 7c*). On the other hand, Rtf1's Plus3 domain was essential for supporting the formation of cardiac progenitors (0% of Rtf1 ΔPlus3 *rtf1* morphants with *nkx2.5* signal, n=25) (*Figure 7c*). These data suggest that Rtf1's domains function in a modular fashion for regulating cardiac gene expression and point to potential specific effects on transcriptional pausing and elongation.

## Rtf1 deficiency disrupts promoter-proximal transcriptional pausing

To examine the occupancy of RNA Pol II, we performed ChIP-seq on embryos at the 10–12 somite stage, when cardiac progenitors are arising from the lateral plate mesoderm. We compared the occupancy of Pol II between uninjected control and *rtf1* morphant embryos and found that Pol II was substantially diminished at the transcription start site (TSS) of many genes compared to the remainder of the gene body (*Figure 8—figure supplement 1* and data not shown). Pausing of RNA Pol II at the TSS is a normal aspect of transcription in vertebrates and has been hypothesized to regulate both the timing and expression level of genes (*Core and Adelman, 2019*; *Adelman and Lis, 2012*). To quantify transcriptional pausing at each gene, we calculated the pause release ratio (PRR), a ratio of Pol II occupancy in the promoter proximal region of a gene (–300 to +300) to the

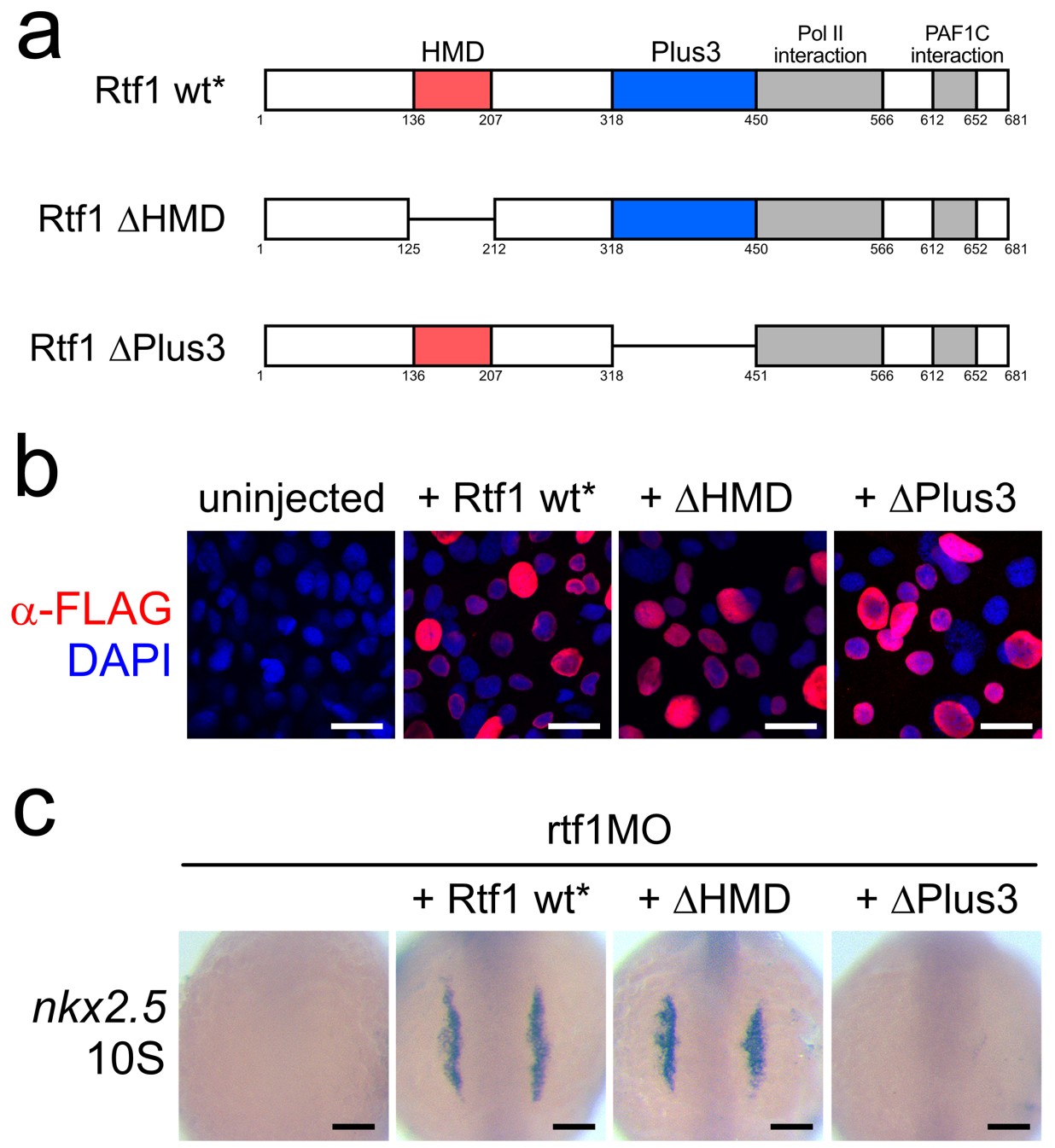

**Figure 7.** Rtf1's Plus3 domain is necessary for cardiac progenitor formation. (**a**) Schematics of Rtf1 wild-type (wt*) and Rtf1 mutant constructs (ΔHMD and ΔPlus3). Domains are indicated by colored boxes (histone modification domain [HMD]: red, Plus3: blue, polymerase II [Pol II], and Polymerase Associated Factor 1 Complex [PAF1C] interaction: gray). Deleted regions are represented by lines. Numbers refer to the amino acid positions in wild-type Rtf1. Rtf1 wt*, ΔHMD, and ΔPlus3 also harbor 8 silent mutations that alter the *rtf1* translation blocking morpholino binding site. (**b**) Projections of confocal z-stacks of whole-mount immunostaining detecting N-terminal FLAG-tagged Rtf1 constructs (red) expressed in 75% epiboly zebrafish embryos. Nuclei are labeled by DAPI staining (blue). Scale bars represent 20 μm. (**c**) Representative images of RNA in situ hybridization detecting *nkx2.5* expression in 10 somite stage (10S) zebrafish embryos co-injected with *rtf1* morpholino and mRNA encoding Rtf1 wild-type (wt*) or mutant (ΔHMD and ΔPlus3) proteins. Scale bars represent 0.1 mm.

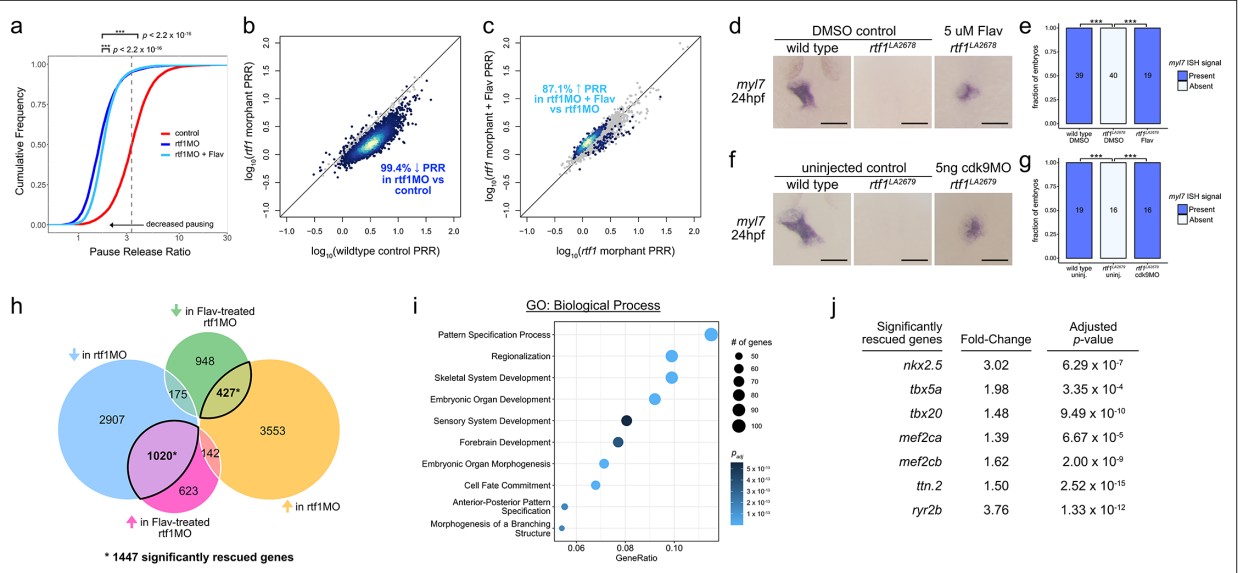

**Figure 8.** Rtf1-dependent transcriptional pausing regulates cardiogenesis. (**a**) Cumulative frequency plot of pause release ratios (PRRs) of 6078 genes with substantial RNA Polymerase II (RNA Pol II) signal. Both control (red) and flavopiridol-treated *rtf1* morphant (light blue) samples displayed significantly different PRR frequencies compared to untreated *rtf1* morphants (blue). ***: p<2.2 × 10⁻¹⁶; Welch's paired two-tailed *t*-test. The median PRR of control samples is indicated by a vertical gray dashed line. (**b**) Dot plot comparing PRRs in controls and *rtf1* morphants. Each dot represents the PRR values for a single gene that differ significantly (colored point) or are not significantly different (gray point) between controls and *rtf1* morphants. (**c**) Dot plot comparing PRRs in *rtf1* morphants and flavopiridol-treated *rtf1* morphants. Each dot represents the PRR values for a single gene that differ significantly (colored point) or are not significantly different (gray point) between *rtf1* morphants and flavopiridol-treated *rtf1* morphants. Dot colors in (**b**) and (**c**) are based on the density of points, with lighter colors indicating more dense points. (**d**) Representative images of RNA in situ hybridization detecting *myl7* expression in 24 hpf control and Rtf1-deficient (±flavopiridol) zebrafish embryos. (**e**) Quantification of *myl7* signal intensity in control and Rtf1-deficient (±flavopiridol) embryos at 24 hpf. Numbers on bars indicate the number of embryos analyzed. ***: p<0.001. (**f**) Representative images of RNA in situ hybridization detecting *myl7* expression in 24 hpf control and Rtf1-deficient (±*cdk9* morpholino) zebrafish embryos. (**g**) Quantification of *myl7* signal intensity in control and Rtf1-deficient (±*cdk9* morpholino) embryos at 24 hpf. Numbers on bars indicate the number of embryos analyzed. ***: p<0.001. (**h**) Venn diagram of significantly altered gene expression in 8–9 somite stage *rtf1* morphant embryos (±flavopiridol treatment). Shaded circles represent genes that are significantly: downregulated in *rtf1* morphants vs. uninjected controls (blue), upregulated in *rtf1* morphants vs. uninjected controls (yellow), downregulated in flavopiridol-treated *rtf1* morphants vs. vehicle-treated (DMSO) *rtf1* morphants (green), and upregulated in flavopiridol-treated *rtf1* morphants vs. vehicle-treated *rtf1* morphants (pink). Overlapping regions with thick black outlines represent the 1447 genes (*) with expression levels that were significantly rescued in 8–9 somite stage *rtf1* morphants by flavopiridol treatment. (**i**) Plot of enriched Biological Process gene ontology terms for the set of genes that were significantly rescued in *rtf1* morphants by flavopiridol treatment. Dot sizes indicate the number of genes associated with a given ontology term. Dot color indicates the significance level (adjusted p-value). (**j**) Selected list of genes that are critical to cardiac development and were significantly rescued in *rtf1* morphants by flavopiridol treatment. Fold-changes and adjusted p-values were calculated with DESeq2. Scale bars in panels d and f represent 0.1 mm.

The online version of this article includes the following figure supplement(s) for figure 8:

**Figure supplement 1.** Promoter-proximal pausing of RNA Polymerase II is diminished at some cardiac mesoderm-related genes in Rtf1-deficient embryos.

occupancy in the downstream region (+301 to +1301). A higher PRR indicates increased pausing of Pol II near the TSS, while a lower PRR reflects a gene in a less paused state. We found that on a genome-wide scale, pausing is diminished upon loss of Rtf1 activity during embryonic development (*Figure 8a and b*). Out of 6078 genes with substantial Pol II ChIP signal that were analyzed, 5799 displayed a significantly decreased PRR (Benjamini-Hochberg false discovery rate of 10%) in Rtf1-deficient embryos compared to 37 genes that displayed a significantly increased PRR (*Figure 8b*). The median PRR in control embryos was 3.34, while 95% (5793/6078) of genes had a PRR of less than 3.34 in Rtf1-deficient embryos (*Figure 8a*), reflecting a widespread decrease in promoter proximally paused Pol II. These changes in PRR suggest that Rtf1 is critical for normal levels of pausing at transcriptionally active genes during the developmental window when cardiac progenitor cells arise from the lateral plate mesoderm.

## Attenuation of pause release permits cardiogenesis in an Rtf1-deficient background

The transition between the pausing and elongation phases of transcription is regulated by PTEF-b, a complex consisting of a cyclin (T1, T2a, or T2b) and the kinase Cdk9 (*Peng et al., 1998*; *Price, 2000*; *Paparidis et al., 2017*). Phosphorylation of the negative elongation factor (NELF) complex, the pausing/elongation factor Spt5, and the C-terminal tail of RNA Pol II by Cdk9 promotes changes in the factors associated with RNA Pol II and triggers a switch to processive elongation (*Fujinaga et al., 2023*). Pharmacological inhibition of Cdk9 with the small-molecule flavopiridol can prevent or attenuate pause release, leading to genome-wide increases in pausing at most genes (*Jonkers et al., 2014*). To explore the effect of Cdk9 inhibition on Rtf1-dependent transcription, we performed ChIP-seq for RNA Pol II in flavopiridol-treated *rtf1* morphant embryos at the 10–12 somite stage. Flavopiridol treatment caused a small but highly significant increase in promoter-proximal pausing in an Rtf1-deficient background (*Figure 8a*), with 87.1% of genes with significantly altered PRR values displaying an increase in PRR in flavopiridol-treated *rtf1* morphants compared to untreated *rtf1* morphants (*Figure 8c*). Furthermore, we found that critical LPM and cardiac genes, including *hand2*, *gata5*, *aplnrb*, *bmp4*, *nkx2.7*, and *rbfox1l,* exhibited reduced promoter-proximal pausing in Rtf1-deficient embryos that was partially rescued by treatment with flavopiridol (*Figure 8—figure supplement 1*).

If diminished promoter-proximal pausing at genes crucial for cardiogenesis is responsible for the absence of cardiac progenitor formation in Rtf1-deficient embryos, then we would predict that reducing pause release may restore cardiogenesis in an Rtf1-deficient background. Indeed, flavopiridol treatment (5 µg/mL dosage) potently restored cardiomyocyte formation in *rtf1* mutant embryos as indicated by *myl7* expression (*Figure 8d and e*). Similarly, knockdown of Cdk9 by injection of antisense morpholino oligonucleotides (*Huang et al., 2022*; *Yang et al., 2016*) also restored the formation of *myl7+* cardiomyocytes in an Rtf1-deficient background (*Figure 8f and g*), confirming that decreased activity of the Cdk9 kinase is responsible for rescuing the *rtf1* cardiac phenotype. These dramatic results suggest that the primary role of Rtf1 in cardiac progenitor formation is regulating promoter-proximal pausing of critical members of the cardiac gene program.

We next performed RNA-seq to examine what transcriptional changes might account for the rescue of *myl7+* cardiomyocyte formation in flavopiridol-treated Rtf1-deficient embryos. Strikingly, we found that of the 4102 genes that were significantly downregulated in *rtf1* morphants at the 8–9 somite stage compared to stage-matched uninjected controls, the expression of 1020 genes was significantly rescued by treatment with flavopiridol (24.9%; *Figure 8h*). Interestingly, we also observed a modest rescue effect on those genes that were significantly upregulated in *rtf1* morphants (427 of 4122 genes significantly rescued: 10.4%; *Figure 8h*). Those genes rescued by flavopiridol treatment were relevant to embryonic development based on significant enrichment for gene ontologies including pattern specification, embryonic organ development, embryonic organ morphogenesis, and cell fate commitment (*Figure 8i*). We examined the expression levels of individual genes critical to cardiac development and found that *nkx2.5*, *tbx5a*, *tbx20*, *mef2ca*, *mef2cb*, *ttn.2*, and *ryr2b* expression levels were all significantly rescued by treatment of Rtf1-deficient embryos with flavopiridol (*Figure 8j*). Altogether, our data support a model in which loss of Rtf1 causes excessive pause release, a dysregulation of the cardiac transcription program, and a failure in the formation of cardiac progenitors.

## Discussion

In this study, we showed that the PAF1C member Rtf1 plays a critical role in cardiogenesis in zebrafish and mammals. Ablation of Rtf1 activity impedes mesoderm differentiation along the myocardial trajectory, resulting in a failure to produce cardiac progenitors with proper expression of cardiac lineage-restricted genes such as *nkx2.5* and *tbx5a*, and the subsequent loss of the cardiomyocyte population. Structure-function analysis revealed that Rtf1's cardiogenic activity requires its Plus3 domain, a site of interaction with the pausing/elongation factor Spt5, and that loss of Rtf1 diminishes promoter-proximal pausing of RNA Pol II. These observations, together with the finding that partial inhibition of transcriptional pause release can support cardiogenesis and restore cardiomyocyte formation in an Rtf1-deficient background, demonstrate a crucial role for Rtf1-mediated transcriptional pausing

in early heart development. These findings also raise an intriguing possibility of potentiating cardiac differentiation through manipulating transcriptional pausing.

How exactly pausing regulates gene expression is not fully understood and likely involves both general and gene/locus-specific mechanisms (*Core and Adelman, 2019*; *Gaertner and Zeitlinger, 2014*). By limiting the release of RNA Pol II into the elongation phase of transcription, pausing can restrain gene expression. This inhibitory role of pausing on transcription allows for another layer of transcription regulation and has been shown to modulate the expression of heat shock and immune stimulus-induced genes (*Gilchrist et al., 2012*; *Adelman et al., 2009*; *Rougvie and Lis, 1988*). Pausing-dependent restraint of gene expression is also important for the timing of gene expression in a developmental context. Supporting this notion, Spt5- and NELF-deficient zebrafish embryos exhibit aberrant upregulation of TGFβ signaling and a resulting inhibition of hematopoiesis (*Yang et al., 2016*). Intriguingly, the PAF1C has also been shown in mammalian cells to restrain transcription from a subset of promoters by inhibiting nearby enhancer activation (*Chen et al., 2017*). On the other hand, pausing also positively regulates gene expression in some contexts. At genes where nucleosome formation is favored in promoter regions, pausing of RNA Pol II can promote a chromatin architecture that is permissive to transcription by competing with nucleosomes for promoter occupancy (*Gilchrist et al., 2010*; *Gilchrist et al., 2008*). This mechanism has been suggested to support expression of the IFN-gamma pathway during zebrafish development (*Yang et al., 2016*). Pausing may also positively regulate gene expression during neural crest development. Loss of Paf1 activity disrupts neural crest differentiation, but this defect can be partially rescued by inactivation of Cdk9. Interestingly, zygotic loss of Cdk9 alone is sufficient to expand the neural crest population in zebrafish embryos, suggesting that promoter-proximal pausing promotes the neural crest gene program (*Jurynec et al., 2019*).

Experiments performed using a rapid protein degradation system in human cell culture have shed light on the mechanisms by which the PAF1C regulates transcriptional pausing and gene expression outputs (*Wang et al., 2022*). In DLD-1 cells, acute depletion of PAF1 destabilizes the PAF1C and stimulates pause release. Loss of PAF1 also decreases chromatin recruitment of INTS11, a component of the Integrator-PP2A complex (INTAC) which interacts with PAF1C and counteracts the activity of P-TEFb. This disruption in the balance of INTAC and P-TEFb activities at transcribed genes results in hyperphosphorylation of SPT5 and likely explains the increase in pause release caused by PAF1 depletion. Similar to our finding that Rtf1 deficiency positively and negatively regulates gene expression, depletion of PAF1 resulted in both upregulation and downregulation of many genes. Intriguingly, the gene expression changes caused by PAF1 depletion correlated with the release frequency and processivity of RNA Pol II at a given gene, with genes exhibiting low levels of release frequency tending to display decreases in their expression level. These observations suggest that the combined regulatory functions of pausing and elongation determine the transcriptional output of PAF1C-regulated genes. Future studies focused on the chromatin architecture of cardiac gene promoters and other regulatory regions, and the dynamics of pause release and elongation at cardiac genes are needed to provide a complete picture of the manner by which promoter-proximal pausing fosters their expression.

## Materials and methods
### Zebrafish
Zebrafish colonies were cared for and bred under standard conditions (*Westerfield, 2000*). Developmental stages of zebrafish embryos were determined based on somite number or hours of development post-fertilization at 28.5°C (*Westerfield, 2000*). Fish husbandry and experiments were performed according to the Institutional Approval for Appropriate Care and Use of Laboratory Animals by the UCLA Institutional Animal Care and Use Committee (Protocols ARC-2000-051-TR-001 and BUA-2018-195-002-CR).

### CRISPR mutagenesis
Template DNA for *rtf1* exon 3 guide RNA (gRNA) (rtf1-e3 gRNA) synthesis was amplified with KOD polymerase (MilliporeSigma) using the pXTW-gRNA plasmid and the primers rtf1-e3-F (5'-TAATACGA CTCACTATAGGAAGAAGGGGAAACCGAGCAAGTTTTAGAGCTAGAAATAGC-3') and gRNA-R (5'-AAAAAAAGCACCGACTCGGTGCCAC-3'). rtf1-e3 gRNA (target sequence: 5'-GGAAGAAGGGGA AACCGAGCAA-3') was synthesized using the MEGAshortscript T7 Transcription Kit (Thermo Fisher

Scientific) and purified using the RNeasy Mini Kit (QIAGEN). Zebrafish embryos were injected at the one-cell stage with 300 pg of Cas9 mRNA and 200 pg of rtf1-e3 gRNA. Injected embryos were raised to adulthood and then outcrossed to wild-type AB fish to establish lines carrying specific *rtf1* mutations.

## Mice

All mice were maintained in the C57BL/6 background according to the Guide for the Care and Use of Laboratory Animals published by the US National Institute of Health. Housing and experiments were performed according to the Institutional Approval for Appropriate Care and Use of Laboratory Animals by the UCLA Institutional Animal Care and Use Committee (Protocols ARC-2020-045-TR-001 and ARC-2020-043-TR-001). ES cells were obtained from KOMP and used to re-derive the *Rtf1*tm1a(-KOMP)Wtsi mouse line. This line was then crossed to a ROSA26::FLPe strain (*Farley et al., 2000*) (The Jackson Laboratory, JAX stock #009086) to create an *Rtf1* conditional knockout allele, where exon 3 of *Rtf1* is flanked by two loxP sites (*Rtf1* flox). These mice were crossed into the *Mesp1*tm2(cre)Ysa background (*Saga et al., 1999*) (generous gift from Dr. Yumiko Saga to AN) to produce a strain capable of generating *Mesp*-Cre+;*Rtf1*flox/flox offspring.

## Whole-mount in situ hybridization

Whole-mount in situ hybridization was performed as described previously (*Chen and Fishman, 1996*). Embryos for in situ hybridization were fixed in 4% paraformaldehyde in 1× PBS. After a brief proteinase K digestion, embryos were incubated with digoxigenin-labeled antisense RNA probes overnight at 65–70°C. Probes were detected using an anti-digoxigenin antibody conjugated with alkaline phosphatase (Roche 11093274910). The zebrafish antisense RNA probes used in this study include *myl7*, *nkx2.5*, *mef2ca*, *tbx20*, and *tbx5a*. The mouse probes used include *Tbx20* and *Nkx2.5*.

Genotypes of zebrafish embryos were confirmed by PCR with primers:

> rtf1-e3-F: 5'-TACAGTGACATCCACCATGCTG-3'
> rtf1-e3-R: 5'-GTAAATAGGGGAGAAACTAATTGAAG-3'.

Genotypes of mouse embryos were confirmed by PCR with primers:

> Rtf1-loxP-F: 5'-GACTGAGAAGGGAAATCTTTGAAAC-3'
> Rtf1-loxP-R: 5'-CTAGTCCTTGAACAAGTTCAGCT-3'
> Cre-F: 5'-GTAGCTGATGATCCGAATAACTAC-3'
> Cre-R: 5'-ATCCAGGTTACGGATATAGT-3'.

## Constructs

To prevent complementarity between *rtf1* morpholino and mRNA sequences, PCR was used to introduce 8 silent mutations to the morpholino target sequence using pCS2/3XFLAG-rtf1 as a template (*Langenbacher et al., 2011*). The morpholino-resistant wild-type *rtf1* sequence was cloned into pCS2/3XFLAG to produce an N-terminally FLAG-tagged morpholino-resistant rtf1 construct (pCS2/3XFLAG-rtf1-mr). Rtf1 deletion mutant sequences were amplified from pCS2/3XFLAG-rtf1-mr by SOE-PCR to produce Rtf1ΔHMD (missing a.a. 126–211 of Rtf1) and Rtf1ΔPlus3 (missing a.a. 319–450 of Rtf1), which were subsequently cloned into pCS2/3XFLAG to produce N-terminally tagged constructs.

## Microinjections

0.5 ng of *rtf1* morpholino (rtf1MO; 5'-CTTTCCGTTTCTTTACATTCACCAT-3') was injected at the one-cell stage (*Langenbacher et al., 2011*; *Akanuma et al., 2007*) along with 1 ng of *p53* morpholino (5'-GCGCCATTGCTTTGCAAGAATTG-3') to prevent p53-dependent apoptosis (*Robu et al., 2007*). For *cdk9* knockdown rescue experiments, 5 ng of *cdk9* morpholino (cdk9MO; 5'-ACACACAAACATCAAATACTCACCC-3') (*Huang et al., 2022*; *Yang et al., 2016*) was injected into an incross of *rtf1*LA2679 heterozygotes at the one-cell stage. 150–200 pg of *rtf1* or *rtf1* deletion mutant mRNAs were co-injected with *rtf1* and *p53* morpholinos at the one-cell stage for mRNA rescue experiments.

## mESC culture and differentiation

E14TG2a mESCs were obtained from the American Type Culture Collection (ATCC) and were cultured on 0.1% gelatin-covered plates without feeder cells. The feeder-free mESCs were maintained in Dulbecco's modified Eagle's medium (Invitrogen) supplemented with 15% fetal bovine serum (FBS; Invitrogen, USA), 1% non-essential amino acids (Invitrogen), 0.1 mM β-mercaptoethanol (Sigma-Aldrich), 1% (vol/vol) penicillin-streptomycin-glutamine (Thermo Fisher Scientific), and 10 μg/μL recombinant mouse leukemia inhibitory factor (LIF; Sigma-Aldrich) in a 5% $CO_2$ incubator at 37°C. The ESC culture medium was changed every 2 days. mESCs were subcultured every 3 days using 0.25% trypsin-EDTA (Invitrogen). mESCs were differentiated in hanging drops (600 cells/20 μL) without LIF under non-adherent conditions. Day 5 EBs were plated onto 0.1% gelatin-covered plates. Media was changed every 2 days during adherent culture.

## Lentivirus transduction

To knock down Rtf1 activity in mESCs, Rtf1 or NT (non-target) shRNA lentivirus harboring puromycin resistance was purchased from Sigma-Aldrich. mESCs were transduced with the indicated virus at a multiplicity of infection of 1 and selected in 1.2 μg/mL puromycin. Rtf1 shRNA knockdown efficacy was tested by western blotting analysis that detected Rtf1 protein level. Antibodies used were anti-Rtf1 (Bethyl Labs A301-329A) and anti-β-actin (Sigma A1978). Relative protein levels were determined by densitometry measurements using Adobe Photoshop.

## Quantitative RT-PCR

Total RNA from NT and shRNA-transduced EBs was extracted using TRIzol RNA isolation reagents (Thermo Fisher Scientific). cDNA was synthesized using the iScript cDNA Synthesis Kit from Bio-Rad. Real-time quantitative PCR was carried out using the Roche LightCycler 480 Real-Time PCR System. Primers for qPCR are listed in *Supplementary file 1*.

## Antibody staining

Embryos injected with mRNA encoding FLAG-tagged zebrafish Rtf1 constructs were fixed in 4% PFA in PBS at 75% epiboly. The fixed embryos were incubated in primary antibody (1:50 mouse anti-FLAG M2, F1804, Sigma) in blocking solution (10% goat serum in PBT) for 2 hr at room temperature followed by detection with fluorescent secondary antibody (1:200 anti-mouse IgG-Alexa Fluor 555, A-31570). Nuclei were stained with DAPI, and embryos were embedded in 1% low-melt agarose and imaged on a Zeiss LSM800 confocal microscope.

## Western blotting

Embryos were lysed in Rubinfeld's Lysis Buffer at 1 day post-fertilization as previously described (*Langenbacher et al., 2011*). Two embryo equivalents were loaded per lane of an 8% polyacrylamide gel. Following electrophoresis, proteins were transferred to a nitrocellulose membrane and detected with antibodies against Rtf1 (Bethyl Labs A301-329A, 1:4000) and β-actin (Sigma A1978, 1:5000). ImageJ was used to perform densitometry comparing the levels of Rtf1 protein between samples using β-actin as the control.

## Chemical treatment

Flavopiridol (Cayman Chemicals) was dissolved in DMSO at a stock concentration of 10 mg/mL. Zebrafish embryos were treated with 1.375–5 μg/mL flavopiridol diluted in E3 buffer beginning at 60–70% epiboly. For *rtf1* mutant rescue experiments, genotypes were confirmed using the primers rtf1-e3-F and rtf1-e3-R.

## FACS

10–12 somite stage TgBAC(hand2:EGFP)[pd24] uninjected and *rtf1* morpholino-injected zebrafish embryos (*Yin et al., 2010*) were nutated in 10 mg/mL protease (Protease from *Bacillus licheniformis*, Sigma P5380) in DMEM/F12 media for 30 min at 4°C and pipetted to dissociate cells. Debris was removed by filtering through a 40 μm cell strainer, and cells were then pelleted by centrifugation at 300 rcf for 5 min at 4°C. Cells were resuspended in cold FACS buffer (1× PBS, 2% FBS, 1 mM EDTA) and refiltered immediately prior to sorting on a BD FACSAria II using a 100 μm nozzle. Dead cells

were excluded by staining with 0.1 µg/mL DAPI. Cells from batches of 100 embryos were sorted directly into a mixture of 350 µL Buffer RA1 and 3.5 µL 2-mercaptoethanol from the NucleoSpin RNA kit (Machery-Nagel) for RNA isolation.

## mRNA-seq

RNA was purified from FACS-sorted cells with the NucleoSpin RNA kit (Machery Nagel). Total RNA was extracted from whole embryos using TRIzol RNA isolation reagents (Thermo Fisher Scientific) and then purified with the NucleoSpin RNA kit (Machery Nagel). Approximately 5–10 ng of input RNA was used to produce libraries with the NEBNext Single Cell/Low Input RNA Library Prep Kit for Illumina (NEB) which were sequenced on an Illumina HiSeq 3000 to produce 50 bp single-end reads. Reads from triplicate samples were mapped to the danRer7 genome using Tophat v2.1.1 (*Kim et al., 2013*) with default parameters. Mapped reads were counted with FeatureCounts v2.0.0 (*Liao et al., 2014*), and differential expression analysis was carried out with DESeq2 v1.32.0 (*Love et al., 2014*). Gene ontology enrichment was examined using the R package clusterProfiler (*Yu et al., 2012*) in R v4.1.0. For gene ontology analysis in LPM FACS experiments, the most significantly downregulated genes in *rtf1* morphant LPM were selected as those genes that were downregulated among the 300 most significantly differentially expressed genes based on adjusted p-value. For gene expression analysis of flavopiridol rescue experiments, genes were considered significantly rescued if flavopiridol treatment resulted in a significant expression change in the direction opposite to that caused by *rtf1* knockdown (adjusted p-value<0.1).

## ChIP-seq

Embryos for ChIP were dissociated with 4 mg/mL Collagenase Type IV and 0.25% Trypsin at room temperature with intermittent pipetting. Dissociation was stopped with 10% FBS in DMEM, and cells were washed once with PBS. Fixation and preparation of chromatin was carried out using the Covaris truChIP Chromatin Shearing Kit and 7 min of sonication in a Covaris E220 ultrasonicator using the recommended settings. Immunoprecipitations were carried out as previously described (*Chicas et al., 2010*) on approximately 5 µg of chromatin (minimum 2.5 µg, maximum 8.8 µg) using 5 µL of total RNA Pol II antibody (Cell Signaling D8L4Y). Libraries were prepared using the NEBNext Ultra II DNA Library Kit for Illumina (NEB) and sequenced with an Illumina NovaSeq 6000 system to produce 50 bp paired-end reads. Reads from triplicate samples were mapped to the danRer11 genome using bowtie2 v2.4.2 (*Langmead and Salzberg, 2012*) (options: `--local` -X 1000), and read densities of UCSC RefSeq gene regions were quantified using the package csaw v1.24.3 (*Lun and Smyth, 2016*; *Lun and Smyth, 2014*) in R v4.0.3. For our pausing analyses, we defined the TSS region as –300 bp to +300 bp surrounding the TSS. RNA Pol II ChIP-seq read densities for the TSS regions of genes displayed a bimodal distribution that was assumed to be two partially overlapping distributions of genes with high and low RNA Pol II TSS signal. We examined those genes with a substantial level of RNA Pol II signal in the TSS region of control samples, defined as those genes present in the upper mode of the distribution of log-transformed TSS read densities based on a cutoff for type I error of 10% using the R cutoff package v0.1.0 (https://github.com/choisy/cutoff; *Choisy, 2022*). The read density in the TSS region was divided by the read density in the region extending from +301 to +1301 bp with respect to the TSS to calculate the PRR. ChIP-seq data was visualized in Python 3.10.0 using the plotting tool SparK v2.6.2 (*Kurtenbach and William Harbour, 2019*).

## Nuclei isolation for single-cell sequencing

Nuclei were isolated from two-hundred 11–12 somite stage zebrafish embryos for 10x Genomics Multiome sequencing using recommended procedures with slight modifications. Embryos were deyolked in deyolking buffer (55 mM NaCl, 1.8 mM KCl, 1.25 mM NaHCO₃), and then embryonic cells were partially dissociated by pipetting in DMEM/F12 media (Gibco 21041025). Cells were pelleted and then lysed on ice for 5 min in 100 µL of chilled 0.1× Lysis buffer (10 mM Tris pH 7.4, 10 mM NaCl, 3 mM MgCl₂, 0.01% Tween-20, 0.01% IGEPAL CA-630, 0.001% digitonin, 1% BSA, 1 mM DTT). 1 mL of Wash buffer (10 mM Tris pH 7.4, 10 mM NaCl, 3 mM MgCl₂, 0.1% Tween-20, 1% BSA, 1 mM DTT, 0.1 U/µL Sigma Protector RNase Inhibitor) was then added to the cell lysis, and the nuclei were pelleted by centrifuging at 500×*g* for 5 min at 4°C. Two more washes with 1 mL of Wash buffer were performed. Nuclei were then resuspended in 50 µL of Nuclei Dilution Buffer (1× 10x Genomics Single

Cell Multiome Nuclei Buffer, 1 mM DTT, 1 U/μL Sigma Protector RNase Inhibitor) and filtered with a 40 μm Flowmi cell strainer. Approximately 10,000 nuclei per sample were targeted for capture with the 10x Chromium Single Cell Multiome ATAC+Gene Expression system by the UCLA Technology Center for Genomics & Bioinformatics.

## Single-cell sequencing analysis

10x Chromium Single Cell Multiome ATAC+Gene Expression libraries were prepared by the UCLA Technology Center for Genomics & Bioinformatics following the manufacturer's recommended protocols and sequenced with an Illumina NovaSeq 6000 system. Reads from RNA-seq and ATAC-seq assays were aligned to the zebrafish GRCz11 genome with the Lawson laboratory's improved transcriptome annotation (*Lawson et al., 2020*) using 10x Genomics' Cell Ranger ARC software v2.0.0 with default parameters. Mapped reads from single-cell RNA-seq and ATAC-seq assays were analyzed in R v4.2.2 using Seurat v4.3.0 (*Hao et al., 2021*). Good-quality cells were retained by filtering using Seurat with the following thresholds: 15,000>RNA UMI>2000, Genes >1500, 20,000>ATAC UMI>5000, TSS enrichment score >3.5. Filtered control and *rtf1* morphant cell datasets were integrated using Harmony v0.1.1 (*Korsunsky et al., 2019*), and preliminary dimensional reduction, cell clustering, and marker gene identification were carried out in Seurat. Clusters of cells that exhibited enriched expression of mitochondrial and ribosomal transcript expression levels but lacked enrichment for other genes were considered abnormal cells and were manually removed from the dataset. Following manual filtering, a final dataset of 14,340 control and 13,839 *rtf1* morphant cells was again integrated with Harmony and multimodal dimensional reduction (weighted-nearest neighbor method with RNA and ATAC assays), cell clustering, and marker gene identification were carried out in Seurat. Cluster cell number proportions and gene expression levels were visualized with Seurat and with published code (*Song et al., 2022*). Based on marker gene expression, cells that were members of clusters corresponding to ALPM (*sema3e+*), endothelium (*flt4+*), cardiac precursors (*mef2cb+*), and rostral blood (*spi1b+*) were extracted, independently integrated with Harmony, and dimensional reduction (RNA assay only), cell clustering, and marker gene identification were carried out in Seurat. Pseudotime analysis of Harmony/Seurat cell embeddings and plotting of gene expression over pseudotime was performed with Monocle v3.0 (*Cao et al., 2019*).

## Statistical analysis

Statistical significance for qualitative differences in gene expression in in situ hybridization experiments was determined using Fisher's exact test, with a p-value of less than 0.05 considered to be significant. Statistical significance of gene expression changes in qPCR analyses was determined based on 2 (for *Nkx2-5* and *Afp* expression) or 3 (for *Myh6*, *Nppa*, *Brachyury*, and *Ncam1* expression) independent experiments using a two-tailed Student's *t*-test with a p-value of less than 0.05 considered to be significant. Statistical significance of the difference between beating cardiomyocyte cluster formation in control and Rtf1 shRNA mESCs was determined based on 4 independent experiments using a two-tailed Student's *t*-test with a p-value of less than 0.05 considered to be significant. To compare PRR means between uninjected, *rtf1* morphant, and flavopiridol-treated *rtf1* morphant embryos, mean PRR values were first log-transformed (log10) and then a Welch's paired two-tailed *t*-test was carried out with a p-value of less than 0.05 considered to be significant. Significantly changed PRRs for individual genes were determined based on 3 independent replicates per condition using the rowttests function in the R package genefilter v1.72.1 to perform two-tailed *t*-tests. The p.adjust function in the R stats package v4.0.3 using a Benjamini and Hochberg (false discovery rate) correction for multiple comparisons was then applied with adjusted p-values of less than 0.1 considered to be significant.

## Acknowledgements

We thank the UCLA Technology Center for Genomics & Bioinformatics for technical assistance with Next Generation Sequencing and the Broad Stem Cell Research Center Flow Cytometry Core Facility for technical assistance with cell sorting. *Figures 3a, 4a* and *Figure 5c* were created with BioRender.com. This work was supported by funding from the NIH/NHLBI to J-NC (R01HL155905 and R01HL140472).

## Additional information

### Funding

| Funder | Grant reference number | Author |
|---|---|---|
| National Institutes of Health | R01 HL140472 | Jau-Nian Chen |
| National Institutes of Health | R01 HL155905 | Jau-Nian Chen |

The funders had no role in study design, data collection and interpretation, or the decision to submit the work for publication.

### Author contributions

Adam D Langenbacher, Conceptualization, Data curation, Formal analysis, Supervision, Validation, Investigation, Visualization, Methodology, Writing - original draft, Writing - review and editing; Fei Lu, Data curation, Formal analysis, Validation, Investigation; Luna Tsang, Zi Yi Stephanie Huang, Formal analysis, Validation; Benjamin Keer, Zhiyu Tian, Alette Eide, Formal analysis; Matteo Pellegrini, Haruko Nakano, Atsushi Nakano, Supervision; Jau-Nian Chen, Conceptualization, Data curation, Formal analysis, Supervision, Funding acquisition, Investigation, Writing - original draft, Project administration, Writing - review and editing

### Author ORCIDs

Adam D Langenbacher  https://orcid.org/0000-0002-3752-0208
Matteo Pellegrini  https://orcid.org/0000-0001-9355-9564
Haruko Nakano  https://orcid.org/0000-0001-5807-9127
Atsushi Nakano  https://orcid.org/0000-0001-5702-5039
Jau-Nian Chen  https://orcid.org/0000-0001-8807-3607

### Ethics

This study was performed in strict association with the recommendation in the Guide for the Care and Use of Laboratory Animals of the National Institutes of Health. Fish husbandry and experiments were performed according to the Institutional Approval for Appropriate Care and Use of Laboratory Animals by the UCLA Institutional Animal Care and Use Committee (Protocols ARC-2000-051-TR-001 and BUA-2018-195-002-CR). All mice were maintained in the C57BL/6 background according to the Guide for the Care and Use of Laboratory Animals published by the US National Institute of Health. Housing and experiments were performed according to the Institutional Approval for Appropriate Care and Use of Laboratory Animals by the UCLA Institutional Animal Care and Use Committee (Protocols ARC-2020-045-TR-001 and ARC-2020-043-TR-001).

Reviewer #1 (Public review): https://doi.org/10.7554/eLife.94524.3.sa1
Reviewer #2 (Public review): https://doi.org/10.7554/eLife.94524.3.sa2
Author response https://doi.org/10.7554/eLife.94524.3.sa3

## Additional files

### Supplementary files

Supplementary file 1. Sequences of primers used for quantitative real-time PCR analysis of gene expression in mouse embryonic stem cells.

Supplementary file 2. Predicted identities of cell types belonging to each Seurat cluster identified in analysis of 11–12 somite stage control and *rtf1* morphant embryo integrated multimodal (ATAC+Gene Expression) single-cell sequencing datasets. Predicted identities were based on manual inspection of marker gene expression.

MDAR checklist

Source data 1. DESeq2 results file containing differential gene expression analysis data comparing FACS-isolated hand2:GFP-positive lateral plate mesoderm (LPM) from control and rtf1 morphant 10–12 somite stage embryos.

Source data 2. Significantly restricted marker genes for Seurat cell clusters from integrated multimodal (ATAC+Gene Expression) single-cell sequencing dataset from control and rtf1 morphant embryos at the 11–12 somite stage.

## Data availability

The RNA-seq, ChIP-seq, and single cell-seq data presented in this study are available in the NCBI SRA (BioProject PRJNA1015262).

The following dataset was generated:

| Author(s) | Year | Dataset title | Dataset URL | Database and Identifier |
|---|---|---|---|---|
| Langenbacher AD, Lu F, Tsang L, Huang ZYS, Keer B, Tian Z, Eide A, Pellegrini M, Nakano H, Nakano A, Chen J-N | 2023 | Rtf1-dependent transcriptional pausing regulates cardiogenesis | https://www.ncbi.nlm. nih.gov/bioproject/? term=PRJNA1015262 | NCBI BioProject, PRJNA1015262 |

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
