## [Editor Report · eLife Assessment]

This **important** study shows that a controlled pause in gene reading is required for early heart cells to form during development. The authors demonstrate that loss of this pause prevents the proper activation of the heart-producing program across animal and stem cell systems. The evidence is **compelling**, supported by careful genomic and functional analyses that clearly define the developmental block. Overall, this work will interest developmental biologists and inspire further studies on the origins of early heart defects.

---

## [Referee Report · Reviewer #1 (Public review)]

This is a highly original and impactful study that significantly advances our understanding of transcriptional regulation, in particular RNAPII pausing, during early heart development. The Chen lab has a long history of producing influential studies in cardiac morphogenesis, and this manuscript represents another thorough and mechanistically insightful contribution. The authors have thoroughly addressed this Reviewer's concerns and incorporated all of my suggestions in the revised manuscript. In addition, their responses to the other reviewer's comments are also very clear. As it is, this work is of great interest to the readership of Elife, as well as to the general scientific community.

The authors reveal a fundamentally new role for Rtf1-a component of the PAF1 complex-in governing promoter-proximal RNAPII pausing in the context of myocardial lineage specification. While transcriptional pausing has been implicated in stress responses and inducible gene programs, its developmental relevance has remained poorly defined. This study fills that gap with rigorous in vivo evidence demonstrating that Rtf1-dependent pausing is indispensable for activating the cardiac gene program from the lateral plate mesoderm.

Importantly, the study also provides compelling therapeutic implications. Showing that CDK9 inhibition-using either flavopiridol or targeted knockdown-can restore promoter-proximal pausing and rescue cardiomyocyte formation in Rtf1-deficient embryos suggests that modulation of pause-release kinetics may represent a new avenue for correcting transcriptionally driven congenital heart defects. Given that many CDK inhibitors are clinically approved or in active development, this connection significantly elevates the translational impact of the findings.

In sum, this study is rigorous, innovative, and transformative in its implications for developmental biology and cardiac medicine. I strongly support its publication.

---

## [Referee Report · Reviewer #2 (Public review)]

Summary:

Langenbacher at el. examine the requirement of Rtf1, a component of the PAF1C complex, which regulates transcriptional pausing in cardiac development. The authors first confirm that newly generated rtf1 mutant alleles recapitulate the defects in cardiac progenitor differentiation found using morpholinos from their previous work. The authors then show that conditional loss of Rtf1 in mouse embryos and depletion in mouse ESCs both demonstrates a failure to turn on cardiac progenitor and differentiation marker genes, supporting conservation of Rtf1 in promoting vertebrate cardiac progenitor development. The authors then employ bulk RNA-seq on flow-sorted hand2:GFP+ cells and multiomic single-cell RNA-seq on whole Rtf1-depleted zebrafish embryos at the 10-12 somite stage. These experiments corroborate that gene expression associated with cardiac progenitor differentiation is lost. Furthermore, analysis of differentiation trajectories suggests that the expression of genes associated with cardiac, blood, and endothelial progenitor differentiation is not initiated within the anterior lateral plate mesoderm. Structure-function analysis supports that the Rtf1 Plus3 domain is necessary for its function in promoting cardiac progenitor differentiation. ChIP-seq for RNA Pol II on 10-12 somite stage zebrafish embryos supports that Rtf1 is required for proper promoter pausing at the transcriptional start site. The transcriptional promoter pausing defect and cardiac differentiation can partially be rescued in zebrafish rtf1 mutants through pharmacological inhibition and depletion of Cdk9, a kinase that inhibits elongation. Thus, the authors have provided a clear analysis of the requirements and basic mechanism that Rf1 employs regulating cardiac progenitor development.

Strengths and weaknesses:

Overall, the data presented are strong and the message of the study is clear. The conclusions that Rtf1 is required for transcriptional pause release and promotes vertebrate cardiac progenitor differentiation are supported. Areas of strength include the complementary approaches in zebrafish and mouse embryos, and mouse embryonic stem cells, which together support the conserved requirement for Rtf1 in promoting cardiac differentiation. The bulk and single-cell RNA-sequencing analyses provide further support for this model via examining broader gene expression. In particular, the pseudotime analysis bolsters that there is a broader effect on differentiation of anterior lateral plate mesoderm derivatives. The structure-function analysis provides a relatively clean demonstration of the requirement of the Rtf1 Plus3 domain. The pharmacological and depletion epistasis of Cdk9 combined with the RNA Pol II ChIP-seq nicely support the mechanism implicating Cdk9 in the Rtf1-dependent RNA Pol II promoter pausing. Additionally, this is a revised manuscript. The authors were overall responsive to the previous critiques. The new analysis and revisions have helped to strengthen their hypothesis and improve the clarity of their study. While the revised manuscript is significantly improved, the lack of analysis from the multiomic analysis still represents a lost opportunity to provide further insight into Rtf1 mechanisms within this study. However, the authors have nevertheless achieved their goal for this study. The data sets reported will also be useful tools for further analysis and integration by the cardiovascular development community. Thus, the study will be of interest to scientists studying cardiovascular development and those broadly interested in epigenetic regulation controlling vertebrate development.

---

## [Author Response]

The following is the authors’ response to the original reviews.

**Public Reviews:**

**Reviewer #1 (Public Review):**
Summary:The manuscript submitted by Langenbacher et al., entitled " Rtf1-dependent transcriptional pausing regulates cardiogenesis", describes very interesting and highly impactful observations about the function of Rtf-1 in cardiac development. Over the last few years, the Chen lab has published novel insights into the genes involved in cardiac morphogenesis. Here, they used the mouse model, the zebrafish model, cellular assays, single cell transcription, chemical inhibition, and pathway analysis to provide a comprehensive view of Rtf1 in RNAPII (Pol2) transcription pausing during cardiac development. They also conducted knockdown-rescue experiments to dissect the functions of Rtf1 domains.Strengths:The most interesting discovery is the connection between Rtf1 and CDK9 in regulating Pol2 pausing as an essential step in normal heart development. The design and execution of these experiments also demonstrate a thorough approach to revealing a previously underappreciated role of Pol2 transcription pausing in cardiac development. This study also highlights the potential amelioration of related cardiac deficiencies using small molecule inhibitors against cyclin dependent kinases, many of which are already clinically approved, while many other specific inhibitors are at various preclinical stages of development for the treatment of other human diseases. Thus, this work is impactful and highly significant.

We thank the reviewer for appreciating our work.

**Reviewer #2 (Public Review):**
Summary:Langenbacher at el. examine the requirement of Rtf1, a component of the PAF1C, which regulates transcriptional pausing in cardiac development. The authors first confirm their previous morphant study with newly generated rtf1 mutant alleles, which recapitulate the defects in cardiac progenitor and diUerentiation gene expression observed previously in morphants. They then examine the conservation of Rtf1 in mouse embryos and embryonic stem cell-derived cardiomyocytes. Conditional loss of Rtf1 in mesodermal lineages and depletion in murine ESCs demonstrates a failure to turn on cardiac progenitor and diUerentiation marker genes, supporting conservation of Rtf1 in promoting cardiac development. The authors subsequently employ bulk RNA-seq on flow-sorted hand2:GFP+ cells and multiomic single-cell RNA-seq on whole Rtf1-depleted embryos at the 10-12 stage. These experiments corroborate that genes associated with cardiac and muscle development are lost. Furthermore, the diUerentiation trajectories suggest that the expression of genes associated with cardiac maturation is not initiated. Structure-function analysis supports that the Plus3 domain is necessary for its function in promoting cardiac progenitor formation. ChIP-seq for RNA Pol II on 1012 somite stage embryos suggests that Rtf1 is required for proper promoter pausing. This defect can partially be rescued through use of a pharmacological inhibitor for Cdk9, which inhibits elongation, can partially restore elongation in rtf1 mutants.Strengths:Many aspects of the data are strong, which support the basic conclusions of the authors that Rtf1 is required for transcriptional pausing and has a conserved requirement in vertebrate cardiac development. Areas of strength include the genetic data supporting the conserved requirement for Rtf1 in promoting cardiac development, the complementary bulk and single-cell RNA-sequencing approaches providing some insight into the gene expression changes of the cardiac progenitors, the structure-function analysis supporting the requirement of the Plus3 domain, and the pharmacological epistasis combined with the RNA Pol II ChIP-seq, supporting the mechanism implicating Cdk9 in the Rtf1 dependent mechanism of RNA Pol II pausing.

We thank the reviewer for the summary and for recognizing many strengths of our work.

Weaknesses:While most of the basic conclusions are supported by the data, there are a number of analyses that are confusing as to why they chose to perform the experiments the way they did and some places where the interpretations presently do not support the interpretations. One of the conclusions is that the phenotype aUects the maturation of the cardiomyocytes and they are arresting in an immature state. However, this seems to be mostly derived from picking a few candidates from the single cell data in Fig. 6. If that were the case, wouldn't the expectation be to observe relatively normal expression of earlier marker genes required for specification, such as Nkx2.5 and Gata5/6? The in situ expression analysis from fish and mice (Fig. 2 and Fig. 3) and bulk RNA-seq (Fig. 5) seems to suggest that there are pretty early specification and diUerentiation defects. While some genes associated with cardiac development are not changed, many of these are not specific to cardiomyocyte progenitors and expressed broadly throughout the ALPM. Similarly, it is not clear why a consistent set of cardiac progenitor genes (for instance mef2ca, nkx2.5, and tbx20) was analyzed for all the experiments, in particular with the single cell analysis.

A major conclusion of our study is that Rtf1 deficiency impairs myocardial lineage differentiation from mesoderm, as suggested by the reviewer. Thus, the main goal of this study is to understand how Rtf1 drives cardiac differentiation from the LPM, rather than the maturation of cardiomyocytes. Multiple lines of evidence support this conclusion:

(a) In situ hybridization showed that Rtf1 mutant embryos do not have nkx2.5+ cardiac progenitor cells and subsequently fail to produce cardiomyocytes (Figs. 2, 3).

(b) RT-PCR analysis showed that knockdown of Rtf1 in mouse embryonic stem cells causes a dramatic reduction of cardiac gene expression and production of significantly fewer beating patches (Fig.4).

(c) Bulk RNA sequencing revealed significant downregulation of cardiac lineage genes, including nkx2.5 (Fig. 5).

(d) Single cell RNA sequencing clearly showed that lateral plate mesoderm (LPM) cells are significantly more abundant in Rtf1 morphant,s whereas cardiac progenitors are less abundant (Fig. 6 and Fig.6 Supplement 1-5).

When feasible, we used cardiac lineage restricted markers in our assays. Nkx2.5 and tbx5a are not highlighted in the single cell analysis because their expression in our sc-seq dataset was too low to examine in the clustering/trajectory analysis. In this revised manuscript, we provide violin plots showing the low expression levels of these genes in single cells from Rtf1 deficient embryos (Figure 6 Supplement 5).

The point of the multiomic analysis is confusing. RNA- and ATAC-seq were apparently done at the same time. Yet, the focus of the analysis that is presented is on a small part of the RNA-seq data. This data set could have been more thoroughly analyzed, particularly in light of how chromatin changes may be associated with the transcriptional pausing. This seems to be a lost opportunity. Additionally, how the single cell data is covered in Supplemental Fig. 2 and 3 is confusing. There is no indication of what the diUerent clusters are in the Figure or the legend.

In this study, we performed single cell multiome analysis and used both scRNAseq and scATACseq datasets to generate reliable clustering. The scRNAseq analysis reveals how Rtf1 deficiency impacts cardiac differentiation from mesoderm, which inspired us to investigate the underlying mechanism and led to the discovery of defects in Rtf1-dependent transcriptional pause release.

We agree with the reviewer that deep examination of Rtf1-dependent chromatin changes would provide additional insights into how Rtf1 influences early development and careful examination of the scATACseq dataset is certainly a good future direction.

In this revised manuscript, we have revised Fig.6 Supplement 1 to include the predicted cell types and provide an additional excel file showing the annotation of all 39 clusters (Supplementary Table 2).

While the effect of Rtf1 loss on cardiomyocyte markers is certainly dramatic, it is not clear how well the mutant fish have been analyzed and how specific the eUect is to this population. It is interpreted that the eUects on cardiomyocytes are not due to "transfating" of other cell fates, yet supplemental Fig. 4 shows numerous eUects on potentially adjacent cell populations. Minimally, additional data needs to be provided showing the live fish at these stages and marker analysis to support these statements. In some images, it is not clear the embryos are the same stage (one can see pigmentation in the eyes of controls that is not in the mutants/morphants), causing some concern about developmental delay in the mutants.

Single cell RNA sequencing showed an increased abundance of LPM cells and a reduced abundance of cardiac progenitors in Rtf1 morphants (Fig. 6 and Fig.6 Supplement 1-5). The reclustering of anterior lateral plate mesoderm (ALPM) cells and their derivatives further showed that cells representing undifferentiated ALPM were increased whereas cells representing all three ALPM derivatives were reduced. These findings indicate a defect in ALPM differentiation.

The reviewer questioned whether we examined stage-matched embryos. In our assay, Rtf1 mutant embryos were collected from crosses of Rtf1 heterozygotes. Each clutch from these crosses consists of ¼ embryos showing rtf1 mutant phenotypes and ¾ embryos showing wild type phenotypes which were used as control. Mutants and their wild type siblings were fixed or analyzed at the same time.

The reviewer questioned the specificity of the Rtf1 deficient cardiac phenotype and pointed out that Rtf1 mutant embryos do not have pigment cells around the eye. Rtf1 is a ubiquitously expressed transcriptional regulator. Previous studies in zebrafish have shown that Rtf1 deficiency significantly impacts embryonic development. Rtf1 deficiency causes severe defects in cardiac lineage and neural crest cell development; consequently, Rtf1 deficient embryos do not have cardiomyocytes and pigmentation (Langenbacher et al., 2011, Akanuma et al., 2007, and Jurynec et al., 2019). We now provide an image showing a 2-day-old Rtf1 mutant embryo and their wild type sibling to illustrate the cardiac, neural crest, and somitogenesis defects caused by loss of Rtf1 activity (Fig. 2 Supplement 1).

With respect to the transcriptional pausing defects in the Rtf1 deficient embryos, it is not clear from the data how this eUect relates to the expression of the cardiac markers. This could have been directly analyzed with some additional sequencing, such as PRO-seq, which would provide a direct analysis of transcriptional elongation.

We showed that Rtf1 deficiency results in a nearly genome-wide decrease in promoterproximal pausing and downregulation of cardiac makers. Attenuating transcriptional pause release could restore cardiomyocyte formation in Rtf1 deficient embryos. In this revised manuscript, we provide additional RNAseq data showing that the expression levels of critical cardiac development genes such as nkx2.5, tbx5a, tbx20, mef2ca, mef2cb, ttn.2, and ryr2b are significantly rescued. We agree with the reviewer that further analyses using the PRO-seq approach could provide additional insights, but it is beyond the scope of this manuscript.

Some additional minor issues include the rationale that sequence conservation suggests an important requirement of a gene (line 137), which there are many examples this isn't the case, referencing figures panels out of order in Figs. 4, 7, and 8 as described in the text, and using the morphants for some experiments, such as the rescue, that could have been done in a blinded manner with the mutants.

We have clarified the rationale in this revised manuscript and made the eRort to reference figures in order.

The reviewer commented that rescue experiments “could have been done in a blinded manner with the mutants”. This was indeed how the flavopiridol rescue and cdk9 knockdown experiments were carried out. Embryos from crosses of Rtf1 heterozygotes were collected, fixed after treatment and subjected to in situ hybridization. Embryos were then scored for cardiac phenotype and genotyped (Fig.8 d-g). Morpholino knockdown was used in genomic experiments because our characterization of rtf1 morphants showed that they faithfully recapitulate the rtf1 mutant phenotype during the timeframe of interest (Fig. 2).

**Recommendations for the authors:**

**Reviewer #1 (Recommendations For The Authors):**
This reviewer has a few suggestions below, aimed at improving the clarity and impact of the current study. Once these items are addressed, the manuscript should be of interest to the Elife reader.Item 1. Strengthening the interaction between Rfh1 and CDK9 on Pol2 pausing.The authors have convincingly shown that the chemical inhibition of CDK9 by flavopiridol can partially rescue the expression of cardiac genes in the zebrafish model. Although flavopiridol is FDA approved and has been a classical inhibitor for the dissection of CDK9 function, it also inhibits related CDKs (such as Flavopiridol (Alvocidib) competes with ATP to inhibit CDKs including CDK1, CDK2, CDK4, CDK6, and CDK9 with IC50 values in the 20-100 nM range) Therefore, this study could be more impactful if the authors can provide evidence on which of these CDKs may be most relevant during Rtf1-dependent cardiogenesis. To determine whether the observed cardiac defect indicates a preferential role for CDK9, or that other CDKs may also be able to provide partial rescue may be clarified using additional, more selective small molecules (e.g., BAY1251152, LDC000067 are commercially available).

The reviewer raised a reasonable concern about the specificity of flavopiridol. We thank the reviewer for the insightful suggestion and share the concern about specificity. To address this question, we have used an orthogonal testing through morpholino inhibition where we directly targeted CDK9 and observed the same level of rescue, supporting a critical role of transcription pausing in cardiogenesis.

Item 2. Differences between CRISPR lines and morphantsMuch of the work presented used Rtf1 morphants while the authors have already generated 2 CRISPR lines. What is the diUerence between morphants and mutants? The authors should comment on the similarities and/or differences between using morphants or mutants in their study and whether the same Rtf1- CDK9 connection also occurs in the CRISPR lines.

The morphology of our mutants (rtf1^LA2678^ and rtf1^LA2679^) resembles the morphants and the previously reported ENU-induced rtf1^KT641^ allele. Extensive in situ hybridization analysis showed that the morphants faithfully recapitulate the mutant phenotypes (Fig.2). We have performed rescue experiments (flavopiridol and CDK9 morpholino) using Rtf1 mutant embryos and found that inhibiting Cdk9 restores cardiomyocyte formation (Fig.8).

Item 3. Discuss the therapeutic relevance of studyThe authors have already generated a mouse model of Rtf1 Mesp1-Cre knockout where cardiac muscle development is severely derailed (Fig 3B). Thus, a demonstration of a conserved role for CDK9 inhibitor in rescuing cardiogenesis using mouse cells or the mouse model will provide important information on a conserved pathway function relevant to mammalian heart development. In the Discussion, how this underlying mechanistic role may be useful in the treatment of congenital heart disease should be provided.

Thank you for the insight. We have incorporated your comments in the discussion.

Item 4. Insights into the role of CDK9-Rtf1 in response to stress versus in cardiogenesis.In the Discussion, the authors commented on the role of additional stress-related stimuli such as heat shock and inflammation that have been linked to CDK9 activity. However, the current ms provides the first, endogenous role of Pol2 pausing in a critical developmental step during normal cardiogenesis. The authors should emphasize the novelty and significance of their work by providing a paragraph on the state of knowledge on the molecular mechanisms governing cardiogenesis, then placing their discovery within this framework. This minor addition will also clarify the significance of this work to the broad readership of eLife.

Thank you for the suggestion. We have incorporated your comments and elaborate on the novelty and significance of our work in the discussion.

**Reviewer #2 (Recommendations For The Authors):**
(1) It is diUicult to assess what the overt defects are in the embryos at any stages. Images of live images were not included in the supplement. Do these have a small, malformed heart tube later or are the embryos just deteriorating due to broad defects?

The Rtf1 deficient embryos do not produce nkx2.5+ cardiac progenitors. Consequently, we never observed a heart tube or detected cells expressing cardiomyocyte marker genes such as myl7. This finding is consistent with previous reports using rtf1 morphants and rtf^1KT64^, an ENU-induced point mutation allele (Langenbacher et al., 2011 and Akanuma, 2007). In this revised manuscript, we provide a live image of 2-day-old wild type and rtf1^LA2679/LA2679^ embryos (Fig. 2 Supplement 1). After two days, rtf1 mutant embryos undergo broad cell death.

(2) Fig. 2, although the in situs are convincing, there is not a quantitative assessment of expression changes for these genes. This could have been done for the bulk or single cell RNA-seq experiments, but was not and these genes weren't not included in the heat maps. A quantitative assessment of these genes would benefit the study.

The top 40 most significantly differentially expressed genes are displayed in the heatmap presented in Fig.5d. The complete differential gene expression analysis results for our hand2 FACS-based comparison of rtf1 morphants and controls is presented in Supplementary Data File 1. In this revised manuscript, we provide a new supplemental figure with violin plots showing the expression levels of genes of interest in our single cell sequencing dataset (Fig.6 Supplement 5).

(3) It doesn't not appear that any statistical tests were used for the comparisons in Fig. 2.

We now provide the statistical data in the legend and Fig.2 b, d, f, h and i.

(4) It's not clear the magnifications and orientations of the embryos in Fig. 3b are the same.

Embryos shown in Fig.3b are at the same magnification. However, because Rtf1 mutant embryos display severe morphological defects, the orientation of mutant embryos was adjusted to examine the cardiac tissue.

(5) The n's for analysis of MLC2v in WT Rtf1 CKO embryos in Fig. 3b are only 1. At least a few more embryos should be analyzed to confirm that the phenotype is consistent.

We have revised the figure and present the number of embryos analyzed and statistics in Fig.3c.

(6) A number of figure panels are referred to out of order in the text. Fig. 4E-G are before Fig. 4C, D, Fig. 7C before 7B, Fig. 8D-I before 8A ,B. In general, it is easier for the reader if the figures panels are presented in the order they are referred to in the text.

Revised as suggested.

(7) While additional genes can be included, it is not clear why the same sets of genes are not examined in the bulk or single-cell RNA-seq as with the in situs or expression was analyzed in embryos. I suggest including the genes like nkx2.5, tbx20, myl7, in all the sequencing analysis.

We used the same set of genes in all analyses when possible. However, the low expression of genes such as nkx2.5 and myl7 in our sc-seq dataset preclude them from the clustering/trajectory analysis. In this revised manuscript, we present violin plots showing their expression in wild type and rtf1 morphants (Fig. 6 Supplement 5).

(8) If a multiomic approach was used, why wasn't its analysis incorporated more into the manuscript? In general, a clearer presentation and deeper analysis of the single cell data would benefit the study. The integration of the RNA and ATAC would benefit the analysis.

As addressed in our response to the reviewer’s public review, both datasets were used in clustering. Examining changes in chromatin accessibility is certainly interesting, but beyond the scope of this study.

(9) Many of the markers analyzed are not cardiac specific or it is not clear they are expressed in cardiac progenitors at the stage of the analysis. Hand2 has broader expression. Additional confirmation of some of the genes through in situ would help the interpretations.

Markers used for the in situ hybridization analysis (myl7, mef2ca, nkx2.5, tbx5a, and tbx20) are known for their critical role in heart development. For sc-seq trajectory analyses, most displayed genes (sema3e, bmp6, ttn.2, mef2cb, tnnt2a, ryr2b, and myh7bb) were identified based on their differential expression along the LPM-cardiac progenitor pseudotime trajectory. Rather than selecting genes based on their cardiac specificity, our goal was to examine the progressive gene expression changes associated with cardiac progenitor formation and compare gene expression of wild type and rtf1 deficient embryos.

(10) Additional labels of the cell clusters are needed for Supplemental Figs. 2 and 3.

The cluster IDs were presented on Supplementary Figures 2 and 3. In this revised version, we added predicted cell types to the UMAP (revised Fig.6 Supplement 1) and provided an excel file with this information (revised Supplementary Table 2).

(11) On lines 101-102, the interpretation from the previous data is that diUerentiation of the LPM requires Rtf1. However, later from the single cell data the interpretation based on the markers is that Rtf1 loss aUects maturation. However, it is not clear this interpretation is correct or what changed from the single cell data. If that were the case, one would expect to see maintenance of more early marks and subsequent loss of maturation markers, which does not appear to the be the case from the presented data.

Our data suggests that cardiac progenitor formation is not accomplished by simultaneously switching on all cardiac marker genes. Our pseudotime trajectory analysis highlights tnnt2a, ryr2b, and myh7bb as genes that increase in expression in a lagged manner compared to mef2cb (Fig. 6). Thus, the abnormal activation of mef2cb without subsequent upregulation of tnnt2a, ryr2b, and myh7bb in rtf1 morphants suggests a requirement for rtf1 in the progressive gene expression changes required for proper cardiac progenitor differentiation. Our single cell experiment focuses on the process of cardiac progenitor differentiation and does not provide insights into cardiomyocyte maturation. We have edited the text to clarify these interpretations.

(12) The interpretation that there is not "transfating" is not supported by the shown data. Analysis of markers in other tissues, again with in situ, to show spatially would benefit the study.

As stated in our response to the reviewer’s public review, we observed a dramatic increase of ALPM cells, but a decrease of ALPM derivatives including the cardiac lineage. We did not observe the expansion of one ALPM-derived subpopulation at the expense of the others. These observations suggest a defect in ALPM differentiation and argue against the notion that the region of the ALPM that would normally give rise to cardiac progenitors is instead differentiating into another cell type.

(13) The rationale that sequence conservation means a gene is important (lines 137-139) is not really true. There are examples a lot of highly conserved genes whose mutants don't have defects.

We have revised the text to avoid confusion.

(14) The data showing that the 8 bp mutations do not aUect the RNA transcript is not shown or at least indicated in Fig. 7. It would seem that this experiment could have been done in the mutant embryos, in which case the experiment would have been semi-blinded as the genotyping would occur after imaging.

The modified Rtf1 wt RNA (Rtf1 wt* in revised Fig. 7) robustly rescued nkx2.5 expression in rtf1 deficient embryos, demonstrating that the 8 bp modifications do not negatively impact the activity of the injected RNA. As stated previously, morpholino knockdown was used in some experiments because our characterization of rtf1 morphants showed that they faithfully recapitulate the rtf1 mutant phenotype during the timeframe of interest.

(15) Using a technique like PRO-seq at the same stage as the ChIP-seq would complement the ChIP-seq and allow a more detailed analysis of the transcriptional pausing on specific genes observed in WT and mutant embryos.

As stated in our response to the reviewer’s public review, we appreciate the suggestion but PRO-seq is beyond the scope of this study.